# Future Indian Ocean warming patterns

**Sahil Sharma** [1,2], **Kyung-Ja Ha** [1,2,3] ✉, **Ryohei Yamaguchi** [4], **Keith B. Rodgers** [1,5], **Axel Timmermann** [1,5] & **Eui-Seok Chung** [6]

Most future projections conducted with coupled general circulation models simulate a non-uniform Indian Ocean warming, with warming hotspots occurring in the Arabian Sea (AS) and the southeastern Indian Ocean (SEIO). But little is known about the underlying physical drivers. Here, we are using a suite of large ensemble simulations of the Community Earth System Model 2 to elucidate the causes of non-uniform Indian Ocean warming. Strong negative air-sea interactions in the Eastern Indian Ocean are responsible for a future weakening of the zonal sea surface temperature gradient, resulting in a slow-down of the Indian Ocean Walker circulation and the generation of south-easterly wind anomalies over the AS. These contribute to anomalous northward ocean heat transport, reduced evaporative cooling, a weakening in upper ocean vertical mixing and an enhanced AS future warming. In contrast, the projected warming in the SEIO is related to a reduction of low-cloud cover and an associated increase in shortwave radiation. Therefore, the regional character of air-sea interactions plays a key role in promoting future large-scale tropical atmospheric circulation anomalies with implications for society and ecosystems far outside the Indian Ocean realm.

Variability in sea surface temperature (SST) over the tropical Indian Ocean (IO) influences mean climate and the variability of the monsoonal precipitation over parts of Asia[1,2], Africa[3], and Australia[4]. Furthermore, IO warming has been suggested to influence North Atlantic SSTs[5], with potential stabilizing effects on the Atlantic meridional overturning circulation[6]. This importance motivated our study to unravel the mechanisms of non-uniform IO warming.

The present-day mean state of the IO is characterized by an eastern basin warm pool, which enhances convection[7] and thereby influences large-scale atmospheric flow, such as the Walker and Hadley circulations. In contrast to the warm pool area, the western IO (WIO) is relatively cold. However, the recent forced changes have caused the IO warm pool region to expand westward[7,8], thereby warming the WIO. Over recent decades, the robust and substantial warming trend observed in the tropical IO has been attributed to changes in atmospheric circulation[9] and associated advection of warm water from the Pacific Ocean via the Indonesian Throughflow[8]. Furthermore, on centennial timescales, the IO warming trend is spatially inhomogeneous in

both observations and climate models, with a more pronounced equatorial WIO warming[10,11]. Over the latter half of the 20th century, the equatorial WIO warming was intensified, even relative to other tropical ocean regions[12]. The warming trend is attributed to an asymmetry in the El Niño–Southern Oscillation teleconnection[10], evaporative damping because of reduced wind speed associated with anomalous easterlies in the central IO[13], and deepening of the thermocline associated with the propagation of downwelling Rossby waves[14]. However, these proposed mechanisms have been found to be inconsistent with each other, and the underlying mechanisms of the warming patterns still remain unclear.

Most future climate model projections, in response to both low and high greenhouse gas emission scenarios, predict substantial mean-state warming in the equatorial WIO relative to that in the rest of the basin[15–19]. Greenhouse gas-induced global warming enhances overall warming over the WIO, thereby weakening the zonal SST gradient (i.e., east minus west), which in turn affects the large-scale monsoonal circulation, and cyclone activity[20], with impacts also on marine primary

[1]Center for Climate Physics, Institute of Basic Science, Busan, South Korea. [2]Department of Climate System, Pusan National University, Busan, South Korea. [3]BK21 School of Earth and Environmental Systems, Pusan National University, Busan, South Korea. [4]Japan Agency for Marine-Earth Science and Technology, Yokosuka, Japan. [5]Pusan National University, Busan, South Korea. [6]Division of Atmospheric Sciences, Korea Polar Research Institute, Incheon, South Korea. ✉e-mail: kjha@pusan.ac.kr

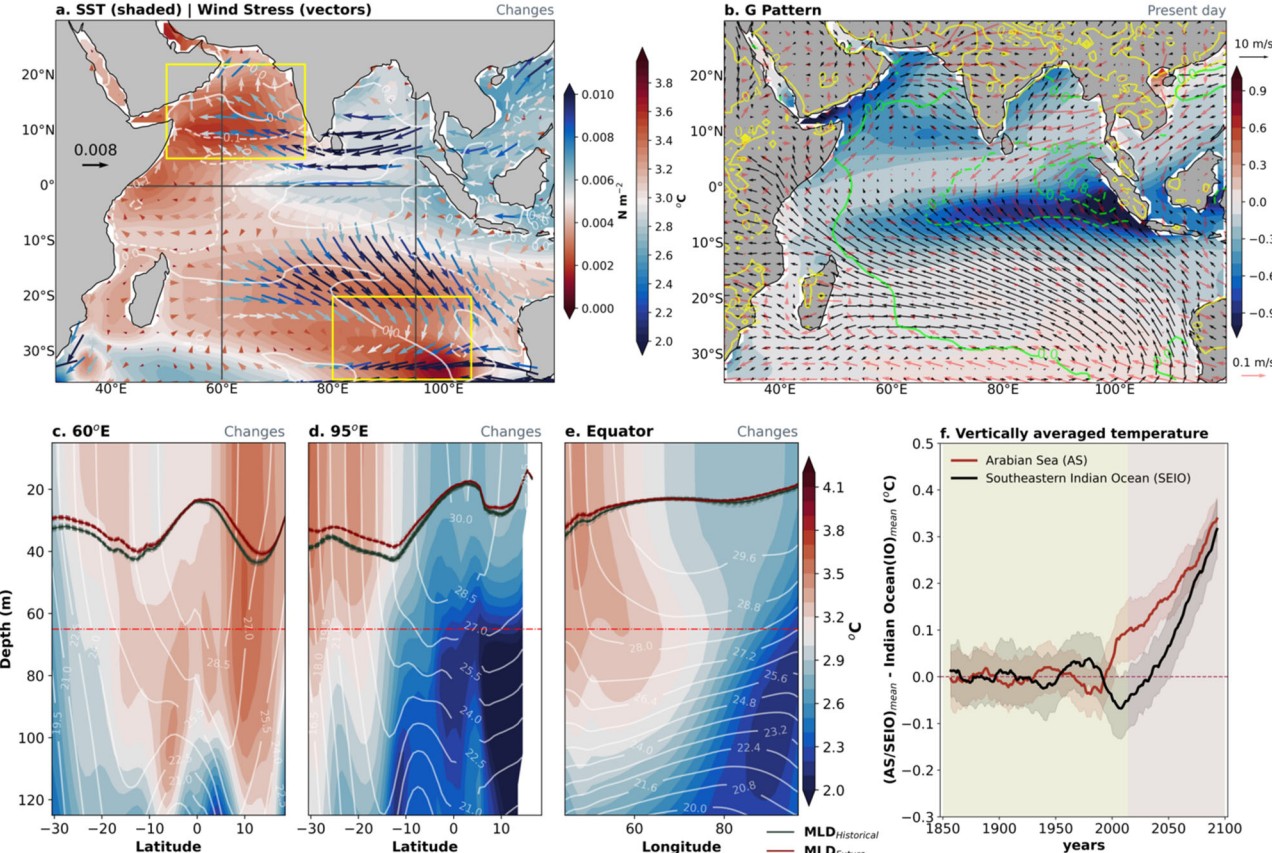

**Fig. 1 | Projected mean-state changes of ocean temperature and present-day air-sea interaction in the Indian Ocean (IO).** Ensemble mean changes of (**a**) sea surface temperature (SST) (°C, shaded) and surface wind stress (N/m², vectors) for the first 50 members of CESM2-LE. The mean state changes are calculated by taking the ensemble mean difference between the future (2080–2100) and the historical (1980–2000) periods. The white contours represent the ensemble standard deviation of the SST changes. The yellow boxes over the Arabian Sea (AS) (5°N:22°N, 50°E:75°E) and southeastern IO (SEIO) (35°S:20°S, 85°E:105°E) indicate the preferential mean-state warming region in the IO in the future climate. Ensemble mean of (**b**) G pattern (shaded), climatological winds (black arrows), anomalous winds (red arrows), and anomalous surface temperature over land (yellow contours) during the period 1980–2000. G is multiplied by a factor of 10³ and anomalies are relative to the 1850–2014 baseline period. The green contours in (**b**) shows the G pattern in the observations during the period 1980–2000 and

anomalies are relative to the 1870–2014 baseline period. The ensemble mean changes in depth-latitude cross-section along (**c**) 60°E, (**d**) 95°E, and (**e**) depth-longitude cross-section along equator (black horizontal and vertical black lines shown in (**a**)). The black and maroon lines along the cross-section indicate the annual-mean mixed layer depth separately during historical and future periods. The white contours represent the historical climatology of subsurface temperature. The red line at 65 m indicates the depth below the mixed layer depth up to which the ocean heat budget analysis is performed. **f** Time series of vertically and area-averaged (0–65 m) temperature over the AS (red) and SEIO (black) during the period 1850 to 2100 for the mean of 50 ensemble members. Note that the IO mean has been removed from each vertically area-averaged time series over the AS and SEIO. Both of the time series are calculated relative to a 1850–2014 baseline, and subsequently a 15-years running has been applied. The shading across the time-series represents the one inter-ensemble standard deviation.

productivity in the Arabian Sea (AS)[21]. Furthermore, the enhanced WIO warming can potentially increase the occurrence and intensity of IO dipole events, as has been observed in recent decades[22], with this trend expected to continue well into the 21st century[15,16,18]. Recent findings have suggested that the response of the IO dipole SST variability is impacted by mean-state changes[18,23,24]. Moreover, enhanced warming over the WIO may trigger Atlantic Niño events by altering the global Walker circulation[25]. Therefore, it is important to elucidate the underlying physical processes that generate the spatial inhomogeneities in forced IO warming.

To better understand the warming patterns in the IO, we analyzed large ensemble simulations conducted with the Community Earth System Model version 2 (CESM2)[26], hereafter referred to as CESM2-Large Ensemble (CESM2-LE)[27] (details are presented in "Methods"). The advantage of large ensemble simulations is that they allow for clear separation between forced climate responses and background natural variability or noise. The CESM2-LE simulations capture the general characteristics of the SST and wind stress in the IO (Supplementary Fig. 1a, b). In addition, the time evolution of SST averaged over the IO,

which shows an increase in recent decades[6,10], is well simulated by the model (Supplementary Fig. 1c).

An analysis of the ensemble mean SST changes shows a weakened zonal equatorial SST gradient (i.e., east minus west), along with enhanced easterlies over the central IO[15–19] (Fig. 1a), consistent with findings in other models[17,28–30]. The projected surface warming is larger in the AS, as compared to the equatorial WIO, as has been pointed out in previous studies. In addition to the intensified warming in the AS, we also see pronounced warming in the south-eastern IO (SEIO) (Fig. 1a, f), again consistent with the findings of other models[31]. Other Coupled Model Intercomparison Project Phase 6 (CMIP6) model simulations exhibit future warming patterns (Supplementary Fig. 2a) which are similar to the CESM2-LE simulations (Fig. 1a). Moreover, the CMIP6 ensemble mean also shows future easterly wind anomalies over the equatorial central IO. In addition, we examined the historical warming pattern in the IO using the observational dataset, which is different, to some degree, from the model-projected future waring pattern (Supplementary Fig. 3). More specifically, the historical warming pattern is characterized by the warming maximum over the equatorial WIO and

the AS along with the secondary local maximum to the south of 20°S. The difference in past and future warming patterns arises because the past warming pattern was shaped mainly by the joint effect natural climate variability and anthropogenic forcings[32,33]. Although it is expected that air-sea interactions are essential for maintaining the mean state, there is a considerable gap in our understanding of how these processes contribute to the projected regional mean state changes, thereby motivating the analysis here.

## Results

### Dynamics of forced warming patterns

Before investigating the controlling factors responsible for the warming pattern in the projected climate using a heat budget analysis, we estimate the present-day air-sea interaction strength ($G = \langle \acute{Q}_{sw} \times \acute{SST} \rangle / \langle \acute{Q}_{sw}^2 \rangle$ where $\acute{Q}_{sw}$ and $\acute{SST}$ represent the shortwave radiative flux and SST perturbation from the climatological mean and the bracket $\langle ... \rangle$ refers to the ensemble mean. Heat flux anomalies in the IO usually serve as a negative feedback for SST anomalies[34,35]. For instance, a positive SST anomaly in this area causes an enhancement of clouds, which in turn reduces shortwave fluxes, and in turn a surface ocean cooling tendency. In addition, we also estimate the atmospheric forcing adjustment processes due to changes in wind dynamics ($Q_{lh}{}^w = \langle \overline{Q_{lh}} \times \acute{W} \rangle / \langle \overline{W} \rangle$; where $\overline{Q_{lh}}$ and $\overline{W}$ are the climatological latent heat flux and surface wind speed and $\acute{W}$ is the perturbation of surface wind speed from the mean). Previous studies highlight the critical role of these two mean air-sea interaction processes in the formation of warming patterns on a global scale[17,36]. Our findings reveal an extended area of negative $G$ in the eastern equatorial IO in both CESM2-LE simulations and observations, which signifies strong atmospheric damping of SST anomalies (Fig. 1b). A positive shortwave flux anomaly leads to an initial surface warming response, which will be damped quickly due to other negative air-sea feedbacks. Translated to the greenhouse warming case, we expect that the regionally enhanced damping (Supplementary Fig. 4) in this area will also counteract longwave radiative changes due to increasing greenhouse gas emissions, which is consistent with the reduced projected warming in this area (Fig. 1a). A reduced future warming in turn will create equatorial easterly wind anomalies, which can generate corresponding shifts in ocean circulation and heat transport, as will be discussed below. Two other key processes need to be considered when linking the eastern IO (EIO) temperature sensitivity with the future warming pattern formation–(i) the projected easterly equatorial wind anomalies shoal the thermocline in the EIO and thereby generate anomalous upwelling of colder subsurface waters. This in turn contributes to offsetting some of the anthropogenic warming, (ii) weak mean wind speeds on the equator reduce the efficiency of the latent cooling process[37], which would enhance the EIO's warming response.

It should be further mentioned that the area of strong damping $G$ in the AS in fact corresponds to a region with increased future warming, which indicates that other processes must overcompensate the strong negative feedbacks there. The enhanced future Arabian land warming causes the anomalous easterlies over the central equatorial IO to bend north-westward over the AS (Supplementary Fig. 5), reducing the mean wind speed and decreasing the latent heat cooling (analogous to $Q_{lh}{}^w$ patterns, Supplementary Fig. 6a) and enhancing the warming response via anomalous oceanic heat transport. To further support our conclusions, we examine the relationship between the variations in wind speed (shortwave radiative flux) and SST over the AS (SEIO). As anticipated from the above explanation, a significant strong negative (positive) correlation of −0.82 (0.69) is observed between the wind speed (shortwave radiative flux) and SST over the AS (SEIO) (Supplementary Fig. 6b, c). These results indicate that the two mean air-sea interaction processes are spatially inhomogeneous, signifying the diversity of underlying regional mechanisms operating for the forced warming patterns in the IO.

To further improve our understanding of the regional warming patterns, we examine projected subsurface ocean temperature changes. The ensemble mean temperature change along 60°E reveals warming that is intensified over the upper-ocean and to the north of the equator (Fig. 1c). The annual mean mixed layer depth structure over the AS is relatively stationary owing to a forced northward shift of the low-level jet stream[38,39]. Similarly, the ensemble mean change for a transect along 95°E reveals near-surface intensified warming and a minor shoaling of the mixed layer depth to the south of 20°S (Fig. 1d). Furthermore, a transect along the equator reveals that the equatorial WIO has stronger near-surface warming than the eastern equatorial IO. (Fig. 1e). To quantitively evaluate the mechanisms underlying distinctive forced upper-ocean warming patterns in the IO, we conducted an ocean heat budget analysis ("Methods"). The ocean heat budget involves heat uptake and release by the ocean, and is primarily governed by four major processes: air-sea heat fluxes, total advection, horizontal diffusion, and ocean interior mixing. The calculations were performed first for each ensemble member before computing the ensemble mean, with control volumes (spanning the surface down to 65 m in the vertical) prescribed separately for both the AS and SEIO, and with the mean for the IO subtracted for each region.

Our analysis indicates that vertical mixing and advection play major roles in anomalous warming over the AS. Horizontal diffusion contributes less, whereas air-sea heat fluxes damp the anomalous warming (Fig. 2a). In addition, the decrease in net surface air-sea fluxes is mainly caused by the reduced downward shortwave radiative flux (Fig. 2a (i)). In order to further deconvolve the drivers, we turn our attention to the individual advection terms to understand how they contribute to the simulated warming patterns. Meridional advection offers a first-order contribution to the tendency, with a minor contribution from zonal advection (Fig. 2a (ii)). We also observe a decreasing contribution of vertical advection over time in response to increasing anthropogenic forcing.

In contrast, perturbed air-sea heat fluxes play a dominant role in the warming over the SEIO, with advection there making only a minor contribution (Fig. 2b). Shortwave fluxes are the largest contributor (Fig. 2b (iii)). Furthermore, diffusive processes (both vertical and horizontal) serve to dampen the warming over the SEIO (Fig. 2b). A decomposition of the advective components reveals only minor contributions there (Fig. 2b (iv)). Thus we see that the processes sustaining future AS and SEIO warming are distinct. Ocean dynamical processes control the former, whereas atmospheric radiative flux perturbations control the latter. The underlying mechanisms that regulate warming in these two regions are discussed in the later sections.

In addition, we elucidate the roles of different processes in the reduced EIO warming in the future. Our analysis indicates that advection, especially zonal advection, among these processes plays a major role in reducing anomalous warming over the EIO (Supplementary Fig. 7a), as was described above. Besides, changes in long wave fluxes associated with decreases in mid- and high-level clouds lead to the reduced EIO warming in the future (Supplementary Fig. 7b).

### Drivers of forced warming pattern over the SEIO

On decadal timescales, the heat balance in the southern IO is mainly regulated by the variability in the Pacific Ocean via the Indonesian throughflow[40–42]. Strong IO warming, observed since 1998, has been closely linked to the Indonesian throughflow[8]. However, future climate change projections indicate a significant reduction in the Indonesian throughflow (Supplementary Fig. 8), consistent with previous findings[43,44], indicating that the southern IO warming is driven by other processes.

As seen in the ocean heat budget analysis, the shortwave radiative flux forcing dominates SEIO warming under future climate change (Fig. 2b). Moreover, the time evolution of air-sea interaction shows a large strengthening towards the end of the 21st century over the SEIO

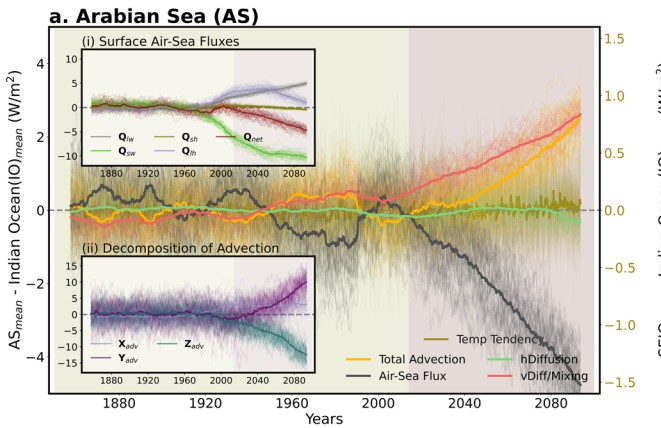

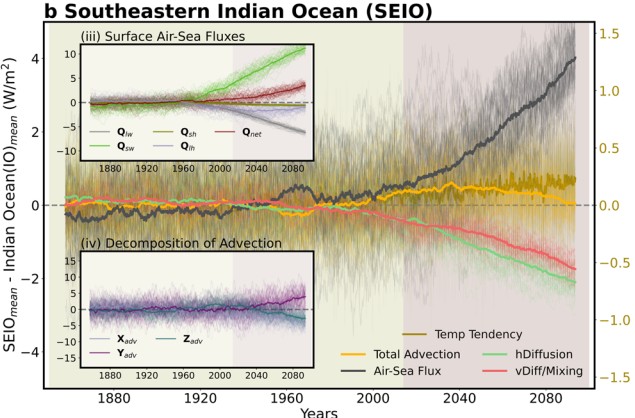

**Fig. 2 | Contribution of different factors to the forced warming pattern.** Time series of each term of vertically integrated (0–65 m) ocean heat budget equation over the **a** Arabian Sea (AS) from 1850–2100 for 50 ensemble members (please see the "Methods" for more details). Inset plot (i) in (**a**) shows the time series of each component of surface air-sea flux (positive values represent anomalous heat input into the ocean) components such as shortwave radiation ($Q_{sw}$), longwave radiation ($Q_{lw}$), sensible heat flux ($Q_{sh}$), latent heat flux ($Q_{lh}$), and net surface air-sea flux ($Q_{net}$)

and (ii) represents the time series of vertically integrated individual advection terms such as zonal ($X_{adv}$), meridional ($Y_{adv}$) and vertical ($Z_{adv}$) advection. Note that IO mean is removed from each area-averaged time series over the AS. All the time series are relative to the 1850–2014 baseline, and finally, the 15-years running mean is calculated. **b** Same as (**a**) but for Southeastern IO (SEIO). Thick (thin) lines represent the ensemble mean (individual ensemble members) in all the plots.

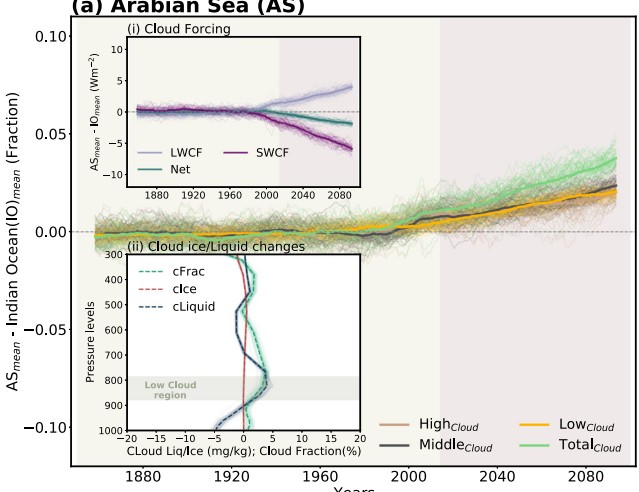

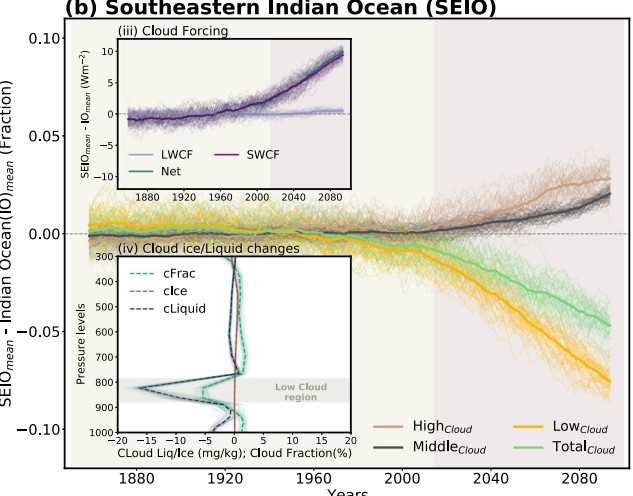

**Fig. 3 | Drivers for future changes in the radiative heat flux over the Arabian Sea (AS) and Southeastern Indian Ocean (SEIO).** Time series of vertically integrated cloud cover over the **a** AS from 1850–2100 for 50 ensemble members. Inset plot (i) in (**a**) shows the time series of the long-wave cloud forcing (LWCF), short-wave cloud forcing (SWCF), and net (LWCF + SWCF) cloud forcing, and (ii) represents the changes in cloud liquid amount, cloud ice amount and cloud fraction over the AS

from the surface to 300 hPa. Note that IO mean is removed from each area-averaged time series over the AS. All the time series are relative to the 1850–2014 baseline, and finally, the 15-years running mean is calculated. **b** Same as (**a**) but for SEIO. Thick (thin) lines represent the ensemble mean (individual ensemble members) in all the plots.

(Supplementary Fig. 9). This demonstrates the important role of the shortwave radiative fluxes for SST changes over the SEIO in a warmer climate. To further support the above results, we also examine the association between the projected changes in $Q_{sw}$ and SST over the SEIO which shows a significant positive correlation ($r = 0.64$) among the ensemble members (Supplementary Fig. 9). Furthermore, we investigate how shortwave radiative flux forcing is related to changes in cloud cover to better understand the mechanisms involved. Clearly, CMIP5 and CMIP6 models exhibit considerable uncertainties in cloud sensitivities. In comparison, CESM2 represents cloud processes and feedbacks in relatively good agreement with observational constraints[45,46] and is thereby deemed to be an appropriate choice for our process-based study. Also, CESM2 produces an excellent representation of the present-day climate system[26,27]. The comparison of present-day time-averaged vertically integrated low and total cloud

cover patterns in the CESM2-LE simulations with the observationally-anchored ERA-5, shows that the CESM2-LE captures well the general characteristics of large-scale pattern of total and low cloud cover (Supplementary Fig. 10).

Initially, we examined the time evolution of cloud radiative forcing (also referred to as the cloud radiative effect), which is characterized by a marked increase in the shortwave component (Fig. 3b (iii)). Total cloud cover over the SEIO is decreased primarily due to a reduction in the low cloud cover (Fig. 3b), resulting in enhanced warming induced by an increase in downward shortwave radiative flux over the ocean surface. The CMIP6 ensemble mean also shows a similar cloud-induced shortwave radiation contribution to anomalous warming over the SEIO in response to greenhouse warming (Supplementary Fig. 2c, d). We further evaluated changes in the vertical distribution of cloud liquid/ice amount along with cloud fraction, in order to better

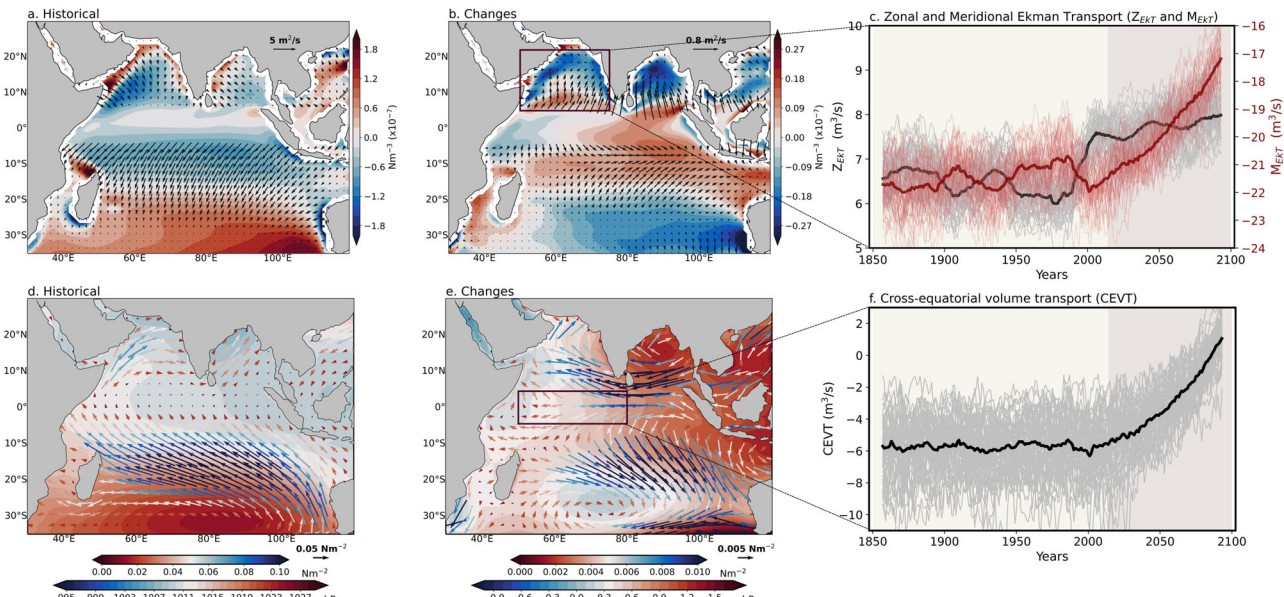

**Fig. 4 | Drivers for future changes in forced warming pattern over Arabian Sea (AS).** Left panel shows the ensemble mean climatology of **a** wind stress curl (shaded) and Ekman transport (vectors), and **d** mean sea level pressure (shaded) and surface wind stress (vectors) for the 50 members of CESM2-LE during the historical period (1980–2000). **b**, **e** Same as in (**a**, **d**) but for the ensemble mean changes by taking the difference between the future (2080–2100) and the historical (1980–2000) periods. The right panel shows area-averaged time series of **c** meridional and zonal Ekman transport over the AS, and **f** cross-equatorial volume transport over the equatorial western Indian Ocean. A 15-year running mean is applied to all the time-series. Thick (thin) lines represent the ensemble mean (individual ensemble members) in all the right panel plots ($Z_{EkT}$ zonal Ekman transport, $M_{EkT}$ meridional Ekman transport, CEVT cross-equatorial volume transport).

understand the forced changes in cloud cover. The ensemble mean shows virtually no change in the cloud ice amount across different pressure levels. In contrast, cloud liquid amount and cloud fraction show large decrease in the lower troposphere, which helps account for the reduction of low-level clouds over the SEIO (Fig. 3b (iv)). In addition, the reduction in low-level clouds is likely to be accompanied by the vertical motion of an anomalous cyclonic circulation (Fig. 1a), which is associated with an increase in mid-level clouds (Fig. 3b) and the weakening of downward motion over the SEIO.

Conversely, over the AS the low-level cloud cover and liquid water amount are projected to increase (Fig. 3a, Fig. 3a (ii)), with the resulting negative shortwave cloud forcing counteracting the positive longwave cloud forcing due to increases in high and mid-cloud cover (Fig. 3a (i)). This demonstrates that the cloud-induced shortwave radiation contribution to warming is minor over the AS in response to 21st century climate change.

**Mechanisms underlying forced warming pattern over the AS**
Our ocean heat budget analysis indicated that meridional advection and ocean interior mixing play important roles in projected AS warming pattern (Fig. 2a). We first investigated ocean circulation changes to determine the physical mechanisms underlying the changes in meridional advection and vertical mixing. Future climate simulations project a weakening of the IO branch of the global Walker circulation, accompanied by larger warming in the equatorial WIO than in the EIO[15,19,28,30], which may strengthen the anomalous easterlies over the central IO (Fig. 1a, Supplementary Fig. 2a). Forced changes in ocean currents include near-surface weakening of the Equatorial Counter Current and strengthening of the North Equatorial Current over the WIO and central IO, respectively (Supplementary Fig. 11). The changes identified in the ocean currents can be related to the weakening of IO Walker circulation in the warmer climate. Considered together, these expected changes may result in the weakening of the southward Ekman transport and an enhancement of anomalous northward Ekman transport over the AS in the future in both CESM2-LE and CMIP6

(Fig. 4a, b, Supplementary Fig. 2b). In addition, the enhanced northward ocean heat transport induces weakening of the cross-equatorial cell, leading to reduced upwelling over the AS in the warmer climate (Supplementary Fig. 11). Moreover, the anomalous anticyclonic wind stress curl over the AS[47] corresponds to anomalous downward Ekman pumping (convergence of heat) and favors forced warming (Fig. 4b).

To further test our conclusions, we examined the contribution of zonal and meridional Ekman transport over the AS. Zonal Ekman transport gradually increases by the end of the 20th century, which is also identified for the zonal advection (Fig. 2a (ii)). Subsequently, however, the transport trend levels off (Fig. 4c). In contrast, the meridional Ekman transport exhibits a gradual increase over the 21st century in both CESM2-LE and CMIP6 (Fig. 4c, Supplementary Fig. 2f), analogous to what was observed for the meridional advection (Fig. 2a (ii)). This finding confirms that the anomalous northward ocean heat transport linked to the slowdown of IO Walker circulation causes an increase in the meridional heat advection, which then contributes predominantly to warming over the AS in the warmer climate.

Furthermore, it is important to determine whether the inhomogeneities in the projected SST trend play a role in the weakening of IO Walker Circulation, or whether the weakening of IO Walker Circulation causes the SST warming pattern. To give an answer, we examined the atmospheric-only model simulations forced by the prescribed SST changes[19,48,49] (see "Methods" for more details). Our analysis reveals that both the uniform and pattern warming experiments exhibit anomalous surface easterly winds over the equatorial central IO. On the other hand, south-easterly wind anomalies shown in coupled model simulations are found only in the uniform warming experiment (Fig. 5a, b). Furthermore, in uniform warming experiment, a multimodel ensemble mean longitude-height cross-section of the vertical velocity indicates a weakening of ascending (descending) motion over the EIO (WIO) (Fig. 5c), representing the slowdown of IO Walker circulation. Moreover, both sets of experiments represent a weakening of the Walker circulation, with the magnitude of anomalous descending

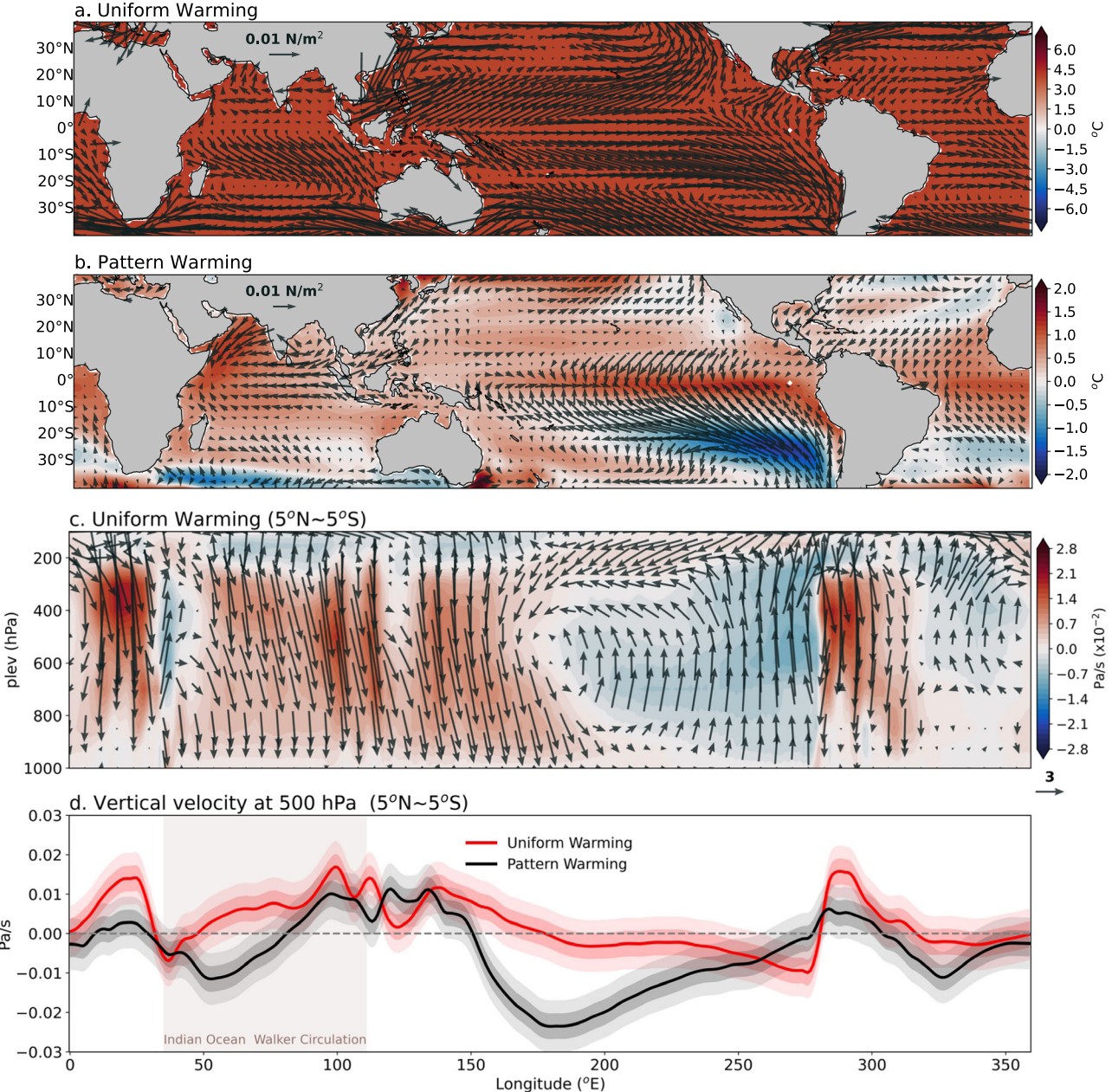

**Fig. 5 | Circulation changes in the uniform and pattern sea surface temperature (SST) warming experiments in atmosphere-only model simulations.** Multi-model ensemble mean changes in surface temperature (shaded, °C) overlain with surface wind-stress (vectors, N/m²) in **a** uniform SST warming experiment, **b** pattern SST warming experiment. **c** The multi-model ensemble mean longitude-height cross-section (5°N-5°S) of vertical velocity (Pa/s, shaded) and zonal-vertical velocity (vectors) from 1000 to 100 hPa in uniform SST warming experiment. The magnitude of vertical velocity is much smaller than zonal velocity, so, vertical velocity is multiplied by the factor 800. **d** Multi-model ensemble mean changes in meridional average (5°N-5°S) of vertical velocity at 500 hPa in the uniform (red) and pattern (black) SST warming experiments. The region of Indian Ocean Walker circulation is shaded in gray. Positive (negative) values in (**c**, **d**) represent an anomalous descending (ascending) motion. The dark (light) shading across the line represents the 0.5 (1) inter-ensemble standard deviation.

motion over the EIO being greater with uniform warming than with pattern warming (Fig. 5d), due to the difference in the tropical-mean or IO-mean warming. This demonstrates that both greenhouse gas-induced warming and SST warming pattern contribute to weakening the IO Walker circulation, with the former exerting somewhat greater influence. Evidence from previous studies has also demonstrated that the IO Walker Circulation can be weakened without the changes in the SST spatial pattern[19,48]. More specifically, they have shown that although the projected weakening of the IO Walker circulation is partly due to the positive IO Dipole-like warming pattern or the El Niño-like warming pattern, the local component in the IO, such as the local meridional circulation, which is independent of the SST spatial pattern,

also plays a crucial role by causing a strong anomalous descending motion over the EIO[19]. This implies that the weakening of Walker Circulation is unlikely to result only from the warming pattern over the equatorial IO. More broadly, as a result of this analysis we can say that the AS warming in response to a weakened IO Walker Circulation is thereby in large part a forced response to the global mean temperature increase.

We also explored an additional physical mechanism that is partly responsible for the enhancement of anomalous northward transport. The climatological pressure gradient between the northern and southern IO drives lower-level southwesterly winds over the AS (Fig. 4d); which further result in southward Ekman transport over the

AS (Fig. 4a). The Mascarene High, a semi-permanent subtropical high-pressure system in the southern IO (Fig. 4d) that modulates the meridional pressure gradient, is responsible for the ocean and atmospheric heat transport over the equatorial WIO[50]. Previous studies have investigated the physical mechanisms that cause the variations in the Mascarene High and their impact on the WIO cross-equatorial winds during the recent global warming hiatus[50]. They showed that enhanced warming over the southern IO is a consequence of the advection of warm water from the western Pacific Ocean via the Indonesian Throughflow. However, the South Equatorial Current is projected to weaken in a warmer climate (Supplementary Fig. 11), reducing the ocean heat transport through the Indonesian Throughflow[43,44] (Supplementary Fig. 8). This indicates that forced warming over the SEIO is not linked to the Pacific Ocean.

Locally induced forced warming over the SEIO not only weakens the intensity of the Mascarene High, but also causes it to shift equatorward, accompanied by a strong cyclonic circulation change (Fig. 4e). The weakening and equatorward migration of the Mascarene High thus weakens the meridional pressure gradient under climate change (Fig. 4d, e) and influences the ocean and atmospheric heat transport over the equatorial WIO[51,52]. The weakening of meridional pressure gradient potentially increases the forced cross-equatorial volume transport over the equatorial WIO (Fig. 4f, Supplementary Fig. 2e) and strengthens the northward transport. This enhances the heat convergence, eventually resulting in enhanced warming over the AS in a warmer climate. As a result, we may conclude that in addition to the weakening of the IO Walker circulation, a future coupling of the intensity of the subtropical high-pressure system with cross-equatorial transport changes over the equatorial WIO may be critical for a better understanding the projected AS warming.

Reduction of cooling by vertical mixing is another primary contributor to anomalous warming over the AS (Fig. 2a). Near-surface density stratification, which regulates the vertical mixing and which has increased over the global domain in the recent decades, is mainly affected by the thermal changes, whereas haline changes are important only locally[53,54]. We further examined the processes that influence future vertical mixing and its contributions to the intensified warming over the AS. Future changes in surface wind speed and SST over the AS are negatively correlated ($r = -0.68$) among the ensemble members, suggesting the reduction in latent heat cooling (Supplementary Fig. 9b, d). In addition, the weakening of the IO Walker circulation shifts the centers of convective activity and induces wetter conditions over the WIO[15,17]. The increase in freshening due to more precipitation (less evaporation) further reduces the surface ocean density (Supplementary Fig. 12) thereby increasing the stability of the water column and enhancing near-surface stratification over the AS in a warmer climate (Supplementary Fig. 13). Interestingly, both the thermal and haline changes contribute equally to the enhanced near-surface stratification. Moreover, in the projected climate scenario, the weakening of the cross-equatorial cell associated with the enhanced anomalous northward ocean heat transport leads to a reduction in upwelling over the AS, further inhibiting the vertical exchange of cold water from the subsurface layers. Thus, the projected enhanced stratification inhibits vertical mixing and thus contributes to forced warming in the AS.

## Discussion

This study examined the forced IO warming patterns using a large ensemble simulation by conducting an extensive ocean heat budget diagnostic. An ocean-atmosphere-based analysis was performed to identify the underlying mechanisms for the IO warming pattern. Our main finding is that ocean circulation changes intensify AS warming, whereas cloud-induced shortwave radiative fluxes control warming in the SEIO.

The main pattern formation mechanism can be summarized as follows: Stronger climatological damping reduces future warming in

the EIO, which contributes to a slowdown of the IO Walker circulation. Corresponding anomalous easterlies over the central IO weaken the upper ocean currents and create an anomalous northward Ekman transport over the AS. Furthermore, the forced weakening and the equatorward shift of the subtropical high-pressure system reduces both the cross-equatorial volume transport over the WIO and upwelling in the AS. Furthermore, near-surface freshening due to increased precipitation and reduced latent heat cooling (due to weakening of mean wind speed caused by enhanced Arabian land warming), decreases the surface ocean density, resulting in enhanced stratification. Increased near-surface stratification reduces vertical mixing and thereby contributes to forced warming patterns in the AS. A schematic diagram of the proposed mechanism of forced warming patterns in the IO is shown in Fig. 6.

Sustained global warming is likely to enhance warming in the AS, with impacts on regional climate and marine biogeochemistry. Moreover, enhanced warming and reduction of mean wind speed in the AS will alter the moisture transport pathways, thereby affecting precipitation in the African and Indian monsoon systems. Understanding how the projected inhomogeneous IO warming will influence rainfall patterns requires a more in-depth analysis of climate model simulations and AMIP-type sensitivity experiments, which is left for future research.

## Methods
### CESM2-large ensemble simulations
This study used large ensemble coupled model simulations conducted using CESM2[26]. The 50 ensemble members are used to study the physical drivers of mean state warming patterns in the IO. The historical and SSP3-7.0 forcing data provided by CMIP6[55] are imposed for all CESM2-LE members, with each ensemble member covering a period from 1850 to 2100. The ocean and atmosphere have a nominal horizontal resolution of 1° × 1°. Specifically, Community Atmosphere Model version 6 (CAM6) has a resolution of 1.25° in longitude and 0.9° in latitude, and 32 vertical levels with a top at 2.26 hPa, or approximately 40 km. The ocean model is the Parallel Ocean Program version 2 (POP2) and have 60 vertical levels in the ocean, the first 20 levels are represented in the upper 200 m of the water column. The periods analyzed in this study spanned 20-years each: 1980–2000 from the historical simulations and 2080–2100 from the projections. We calculate the projected changes by taking the difference between the future and the historical periods for all the ensemble members. For the ocean heat budget analysis, we examined the temporal evolution of individual terms from 1850 to 2100. The key simulated climate indicators in CESM2-LE match closely with the observations during the historical period (see Fig. 1 of the presentation paper[27]). Further details related to the model configuration, large ensemble initialization, and forcing are presented in the CESM2-LE presentation paper[27].

To determine the robustness of our results obtained from the CESM2-LE simulations, we compared them to results from other simulations conducted with CMIP6-CESM2. The annual mean spatial patterns of differences between the idealized abrupt4×CO2 and pre-industrial control experiments in CMIP6-CESM indicate that anomalous warming over the SEIO is associated with the cloud-induced shortwave radiative flux changes, i.e., enhanced downward shortwave radiation owing to reduced low-level clouds (Supplementary Fig. 14), which is consistent with results from CESM2-LE simulations. Furthermore, the simulations using pre-industrial climate and idealized 4×CO2 perturbations in CMIP6-CESM2 reveal that reductions in southward transport are associated with the weakening of cross-equatorial transport and the meridional pressure gradient (Supplementary Fig. 15), which is again consistent with the results from CESM2-LE simulations. To further demonstrate the robustness of the results, we also examined idealized abrupt4×CO2 and pre-industrial

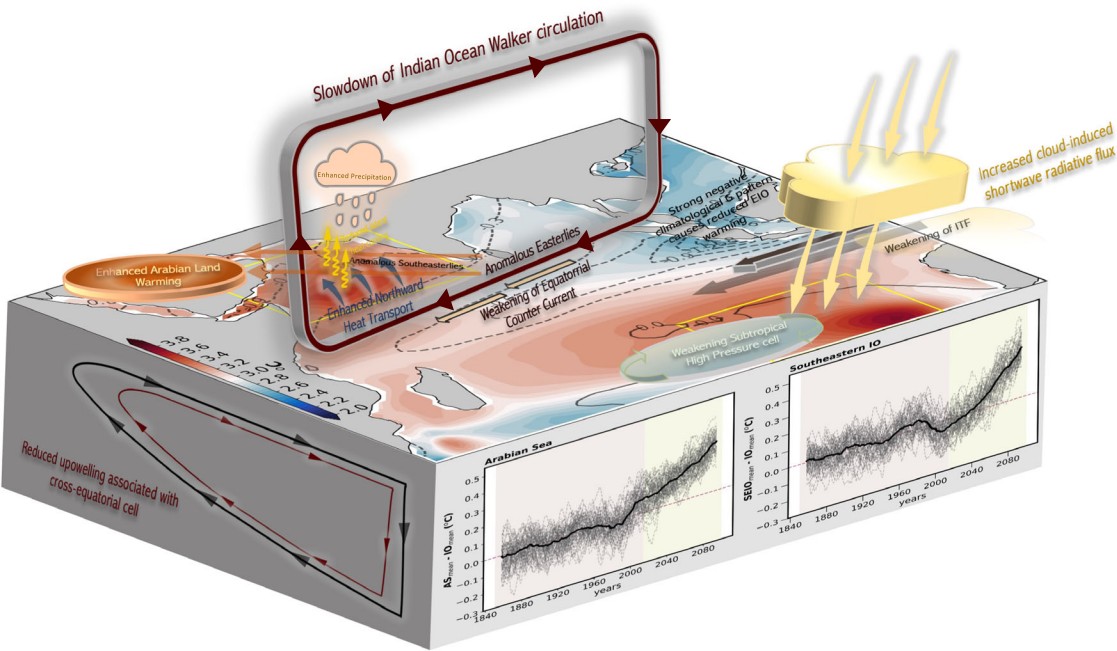

**Fig. 6 | Schematic diagram of the Indian Ocean warming pattern mechanism.**
Ensemble mean of climatological $G$ pattern (contours) during the period
1980–2000 and changes (future minus historical) in SST (shaded) for the 50
members of CESM2-LE. The negative values of $G$ (dashed contours) represent
strong negative feedback and thus a dampening of SST anomalies. A 15-year
running mean time series of the SST from 1850–2100 for 50 ensemble members
(thin) and ensemble mean (thick) over the AS (5°N:22°N, 50°E:75°E) and SEIO
(35°S:20°S, 85°E:105°E). Each of the physical process accountable for forced
warming patterns over the AS and SEIO are emphasized.

control experiments in other CMIP6 models and the multi-model
ensemble mean shows similar reductions in southward transport,
weakening of cross-equatorial transport and the meridional pressure
gradient (Supplementary Fig. 16), consistent with CESM2-LE
simulations.

## Coupled model and atmosphere-only model simulations from CMIP6

We used coupled model simulation outputs from the CMIP6 (17
models) archive for the historical and SSP3-7.0 scenario. The CMIP6
models used for the analysis are BCC-CSM2-MR, CAMS-CSM1-0,
CESM2-FV2, CESM2, CESM2-WACCM, CMCC-ESM2, CanESM5, GISS-E2-
1-G, MCM-UA-1-0, MRI-ESM2-0, MPI-ESM1-2-LR, MIROC-ES2L, MIROC6,
NorESM2-LM, TaiESM1, UKESM1-0-LL, and UKESM1-1-LL. As in the
CESM2-LE, we calculate the projected changes by taking the difference
between the future (2080–2100) and the historical (1980–2000) per-
iods for the multi-model ensemble mean. For each model only first
realization is used in this study.

To determine the cause-and-effect relationship, we analyzed the
atmosphere-only model simulations forced by prescribed SST changes
using AMIP6 simulations available through ESGF. The atmosphere-only
simulations from AMIP6 are (1) AMIP run that uses historical observed
SST and external forcing from 1979 to 2014; (2) AMIPP4K simulations
that are otherwise the same as the AMIP experiments but SST pertur-
bations consist of a uniform warming of 4°K; (3) AMIPFuture4K
simulations that are otherwise the same as the AMIP experiments for
the historical period, except that SSTs are subject to a composite of
SST warming pattern derived from coupled models, scaled to an ice-
free ocean mean of 4K. We have subtracted AMIP from AMIPP4K to get
the response to uniform SST perturbations (uniform warming) in
AMIP6 simulations. Also, to get the response to pattern SST change
(pattern warming), we subtract AMIPP4K from AMIPFuture4K simu-
lations. In the uniform warming experiment, the SST uniformly
increases over the ocean; it does not affect the mean east-west gra-
dient in the base climate. The AMIP6 models used for the analysis are

BCC-CSM2-MR, CNRM-CM6-1, CESM2, CanESM5, E3SM-1-0, HadGEM3-
GC31-LL, IPSL-CM6A-LR, TaiESM1, and MRI-ESM2-0.

## Ocean heat budget analysis

To determine the underlying mechanisms of, as well as contributions
made by, different factors in the anomalous warming pattern in the IO,
we conducted an ocean heat budget analysis[56–58] by integrating the
temperature equation vertically from the surface to a depth of 65 m as
follows:

$$
\int_{-h}^{0} \frac{\partial T}{\partial t}\, dz = \frac{1}{c_p \rho_o}\left(\frac{\partial Q_{net} - Q_{sw}}{\partial z}\right)_{surf} + \frac{1}{c_p \rho_o}\int_{-h}^{0}\frac{\partial Q_{swPen}}{\partial z}\, dz
$$
$$
- \int_{-h}^{0}\nabla.uT\, dz + \int_{-h}^{0} K_h\left(\frac{d^2 T}{dx^2} + \frac{d^2 T}{dy^2}\right) dz \qquad (1)
$$
$$
+ \int_{-h}^{0}\frac{\partial}{\partial z}\left(K_v \frac{\partial T}{\partial z}\right) dz \ldots..
$$

In Eq. (1) above, the term on the left-hand side indicates the
temperature tendency. The first and second term on the right-hand
side denotes the air-sea heat exchange and ocean heating due to the
short-wave penetration. The third term represents the vertical/hor-
izontal advection. The fourth term describes horizontal diffusion,
which includes contributions from the eddy-induced advection, Redi
tendency, Robert filter tendency, and sub-meso scale advection in the
model. The fifth term is diapycnal/KPP vertical mixing in the model.
$\rho_o c_p$ represents seawater density multiplied by the specific heat of
seawater, the value of which is $4.1 \times 10^6$ J/m³/K. $Q_{net}$ is the net heat flux
at the ocean surface, which includes shortwave radiation, longwave
radiation, latent heat flux, and sensible heat flux. $Q_{sw}$ is the shortwave
radiation at the surface and $Q_{swPen}$ is the penetration of solar radiation.

The decomposition of total advection in Eq. (1) into components produces:

$$X_{adv} = -C_p\rho_o \int_{-h}^{0} u\frac{\partial T}{\partial x}dz \ldots\ldots \quad (2)$$

$$Y_{adv} = -C_p\rho_o \int_{-h}^{0} v\frac{\partial T}{\partial y}dz \ldots\ldots \quad (3)$$

$$Z_{adv} = -C_p\rho_o \int_{-h}^{0} w\frac{\partial T}{\partial z}dz \ldots\ldots \quad (4)$$

The decomposition of the total advection produces zonal ($X_{adv}$), meridional ($Y_{adv}$), and vertical ($Z_{adv}$) advection components. In Eqs. (2), (3), and (4) above, u, v, and w represent the zonal, meridional, and vertical ocean velocities, respectively, whereas, $\frac{\partial T}{\partial x}$, $\frac{\partial T}{\partial y}$, and $\frac{\partial T}{\partial z}$ are the zonal, meridional and vertical gradients of temperature at each depth level, respectively. All integration in the ocean heat budget analysis is performed from the surface to a depth of 65 m, thereby defining the upper and lower boundaries of the control volumes. To avoid the effect of very large vertical mixing within the mixed layer, we selected a depth below the mixed layer depth level to perform integration (Fig. 1c). We use the thickness of the turbulent mixing layer calculated by the ocean mixed layer model (KPP scheme) in the CESM2 as the definition of the mixed layer depth. We also found that the pattern of ocean heat content changes within the upper 65 m depth and SST changes are highly consistent with each other (Fig. 1a, Supplementary Fig. 17).

All the ocean heat budget terms are obtained for the time series of 1850–2100. In each time series, the IO mean values are subtracted from the corresponding area-averaged values over the AS and SEIO. All the time series are relative to the 1850–2014 baseline, and the 15-years running mean is finally calculated.

### Other derived variables

We computed the zonal and meridional Ekman Transport over the AS to assess surface transport changes modulated by wind stress. We also analyzed the cross-equatorial volume transport over the equatorial WIO and its role in enhancing the anomalous northward transport.

$$Z_{ekT} = \int_{y_s}^{y_n} \frac{\tau_y}{f\rho_o}dy \ldots\ldots \quad (5)$$

$$M_{ekT} = \int_{X_w}^{X_e} \frac{\tau_x}{f\rho_o}dx \ldots\ldots \quad (6)$$

$$CEVT = \frac{1}{\beta\rho_o} \int_{X_w}^{X_e} \frac{\partial\tau_x}{\partial y}dx \ldots\ldots \quad (7)$$

In Eq. (5) above, $y_n$ (22°N) and $y_s$ (5°N) are the locations of the northern and southern boundaries, respectively. For more details, please see the yellow box over the AS in Fig. 1a. $\tau_y$ is the meridional wind stress and $f = 2\omega\sin\phi$ is the Coriolis parameter. $\rho_o$ represents the density of seawater (1025 kg/m³). In Eq. (6), $X_w$ (50°E) and $X_e$ (75°E) are the locations of the western and eastern boundaries, respectively. $\tau_x$ is the zonal wind stress. In Eq. (7), the cross-equatorial volume transport (CEVT) in the equatorial WIO is obtained by the taking the difference of meridional gradient of the zonal wind stress along 5°N and 5°S (5° transect). $\beta = \frac{df}{dy}$ is the meridional gradient of the Coriolis

parameter. The less negative values of CEVT represent a reduction in the southward transport.

To understand the underlying processes driving the changes in the vertical mixing which contributes to forming the forced warming patterns in the AS, the enhanced near-surface stratification were evaluated. Near-surface density stratification is mainly sensitive to thermal- and haline-driven changes. The relative roles of thermal and haline changes on the density stratification changes are computed using the following equation:

$$\Delta\rho = \alpha\Delta T + \beta\Delta S \ldots\ldots \quad (8)$$

where $\Delta\rho$, $\Delta T$, and $\Delta S$ are the vertical gradients of density, temperature and salinity, respectively. In addition, $\alpha\left(-\frac{1}{\rho}\frac{\partial\rho}{\partial T}\right)$ and $\beta\left(\frac{1}{\rho}\frac{\partial\rho}{\partial S}\right)$ are the thermal expansion and haline contraction coefficients, respectively. To examine near-surface stratification, we analyzed the time evolution of each term at a depth of 65 m.

## Data availability

The CESM2-LE model output is available through (https://www.cesm.ucar.edu/projects/community-projects/LENS2/data-sets.html). The CMIP6-CESM, CMIP6, and AMIP6 models data output for the historical and future period, as well as for the idealized abrupt4×CO2 and pre-industrial control experiments is available through (https://esgf-node.llnl.gov/projects/cmip6). HadISST data can be downloaded from the Met Office Hadley Centre (https://www.metoffice.gov.uk/hadobs/hadisst/). The wind-stress data is available through (https://www2.atmos.umd.edu/~ocean/index_files/soda3.4.1_mn_download.htm). The cloud data is available through (https://www.ecmwf.int/en/forecasts/datasets/reanalysis-datasets/era5). NOAA- 20th Century Reanalysis V3 downward solar radiative flux dataset for the calculation of G pattern is available through (https://psl.noaa.gov/data/20thC_Rean/).

## Code availability

The python code used to run the analysis can be obtained upon request to the corresponding author.

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

## Acknowledgements

S.S., K.-J.H., K.B., and A.T. were supported by the Institute for Basic Science (IBS), Republic of Korea, under grant IBS-RO28-D1. The simulations presented here with CESM2-LE were conducted on the IBS/ICCP supercomputer "Aleph", a 1.43 petaflop high-performance Cray XC50-LC Skylake computing system with 18,720 processor cores, 9.59 PB of disk storage, and 43 PB of tape archive storage. We also acknowledge the support of KREONET. We also acknowledge the World Climate Research Programme's Working Group on Coupled Modeling, which is responsible for the CMIP, and we thank the climate modeling groups for producing and making available their model output.

## Author contributions

S.S. and K.-J.H. designed the study. S.S. performed all the analysis and produced figures with support from R.Y. S.S., R.Y., and K.B.R. initially discussed the scientific results and analyses. K.-J.H., R.Y., K.B.R., A.T., and E.-S.C. provided feedback on the analyses, interpretation of results, and figures. S.S., K.-J.H., R.Y., K.B.R., A.T., and E-S.C. contributed towards the writing of the manuscript.

## Competing interests

The authors declare no competing interests.
