## [Peer Review File · Nature Communications]

Future Indian Ocean warming patternsREVIEWER COMMENTS

Reviewer #1 (Remarks to the Author):

Manuscript Title: Future Indian Ocean warming patterns

Major comments:

1. This paper points out that the future Indian Ocean warming hotspots are tied to the weakening of the IWC. My major concern is that the conclusions are mostly drawn by concurrent composite maps, which obscure the causal chain and are not convincing enough. The author may consider adding sensitivity experiments using CESM2, including more CMIP models for cross-model validations and/or analyzing lead-lag relationships.

2. Calculating the relative roles of thermal and haline changes on stratification is inappropriate because T and S are not mutually independent. There are other ways to compare contributions from the two, such as the Monin-Obukhov (M-O) stability parameter (the ratio of thermocline depth to the Monin-Obukhov length).

Minor comments:

1. Line 81: "strengthening and zonal SST gradient"—>"weakening the zonal SST gradient"?

2. Line 96: how well can the CESM2 simulate the historical IO warming patterns?

3. Line 125: No corresponding Figures for "the enhanced damping...". Fig.1 only shows the climatological G? Suggest showing its future changes as well.

4. Line 343: Have the author looked at other CMIP6 models?

5. Line 375: since the heat budget analyses are applied to the upper 65m below the mixed layer depth, it is necessary to check that the ocean heat content variation within the upper 65m is in high consistency with the sea surface warming patterns.

6. Figure 1: how about the ensemble spread?

7. Figure 2: better to give the time series of Temp because it is hard to see the Temp change with time by the tendency.

8. Figure 5: something wrong with the caption : "G pattern (shaded)" 、 "SST (°C, shaded)". Consider using shadings for SSTs, and contour lines for G.

9. Lines 101-102 "the projected intensified surface warming is confined to the northern WIO", I think such a description is not proper. As seen in Figure 1, there is also intensified surface warming in equatorial WIO. Thinking about rephrasing the description.

10. Lines 208-209, You mention that the reduction in the low-cloud cover is caused by the low-level anomalous anticyclonic circulation over the SEIO. However, I see anomalous cyclonic circulation over the SEIO.

11. Lines 351-352, please double-check the expression and sign of the first and second term on the right-hand side in equation (1).

12. Line 393-395, why is the cross-equatorial volume transport in the WIO calculated on the 5° transect? Does 5° mean 5°N?

13. Line 588, It is better to tell people how you define mixed layer depth.

14. Line 152 in the supplementary, "the slope across the across the ensemble

members", double "across the".

Reviewer #2 (Remarks to the Author):

Review of "Future Indian Ocean warming patterns" by Sharma et al.

This study investigates the formation mechanism of the Indian Ocean warming pattern under global warming by analyzing CESM2 large ensemble simulations. The authors conclude that the weak equatorial Indian Ocean warming is due to a negative cloud-SST feedback, the enhanced southeast Indian Ocean warming is caused by reduced cloudiness, and the strong Arabian Sea warming is attributed to changes in meridional advection and vertical mixing. The Indian Ocean warming pattern formation has been analyzed extensively in the past, and I am afraid that I have not found significant enough contribution from this study that warrants publication in Nature Communications. Besides, I have several concerns about the conclusions, which are related to model biases in simulating clouds and the atmosphere-ocean coupling processes. Based on my evaluation, I don't find this manuscript suitable for publication in its current form. I have listed my comments below.

Specific comments:

- 1. Slowdown of the Indian Ocean Walker cell is a robust feature that can be found in most climate model projections, and as suggested by the authors, it plays an important role in affecting the Indian Ocean warming pattern. This is not too surprising and has been discussed in previous literatures already. However, I think the important question that needs addressed is what causes changes in the Indian Ocean winds. It seems that the authors have attributed the easterly wind anomalies to the weaker EIO warming (L36-40), but since atmosphere and ocean are fully coupled, the wind changes could in turn induce the weaker EIO warming in the first place. In particular, the well-known El Niño-like warming in the tropical Pacific could induce such Indian Ocean wind changes and subsequently affect Indian Ocean warming pattern. Hence, I don't think the analysis in this study could fully support the conclusion that the weaker EIO warming is due to a negative cloud-SST feedback.**
- 2. It seems that cloud changes play an important role in affecting SST changes. However, current generation of climate models still have quite large uncertainties in simulating cloud changes, especially low clouds. How this may affect the conclusions in this study, especially for the southeast Indian Ocean where the reduced low clouds seem crucial in causing SST warming?**
- 3. The wind changes seem important for the AS warming, which have been further attributed to the Arabian land warming (L138-139). I find this argument not quite convincing. For instance, the land warming over India is also very strong, yet we don't see winds converging toward the region. It seems to me that the easterly wind anomalies over the AS could be driven by the strong SST warming instead. I think more evidence is needed to support the argument.**
- 4. Heat budget analysis is crucial for drawing conclusions in this study. However, I notice that the budget is performed with a fixed mixed layer depth (L161-162), which is much deeper than the actual MLD (Fig. 1). I wonder how the results may change if the authors use the MLD in the heat budget instead.**

Reviewer #3 (Remarks to the Author):

Review of the manuscript "Future Indian Ocean warming patterns" by Sharma et al.

Reviewer comments:

This paper examines the mechanism for the non-uniform warming pattern over Indian Ocean. Enhanced warming hotspots are found in the Arabian Sea (AS) and the southeastern Indian Ocean (SEIO). The heat budget analysis is conducted to identify the relative contribution of atmospheric and oceanic processes. The ocean heat transport mainly contributes to the enhanced AS warming, while the cloud-shortwave radiation

plays a key role in the SEIO warming. The manuscript is well written, and the finding is interesting and robust. However, some processes still remain unclear and more evidences are needed. I recommend a major revision before the paper can be published in Nature Communications.

Major comment:

1. The main concern is about the mechanism for the enhanced warming over AS. In this mechanism, the weakened IO Walker circulation is the reason for anomalous heat transport responsible for the AS warming. The question is whether the Walker circulation is induced by the enhanced warming over the WIO or the AS? If so, they are coupled and we cannot say which is the cause. Maybe an experiment with uniform SST warming can answer this question. I suggest to adding an analysis based on a uniform SST warming experiment. If the weakened IO Walker circulation still exists in the uniform SST warming experiment, we can believe the weakened IO Walker circulation is the cause of the enhanced AS warming, and your argument would make sense.

2. Line 81: In my opinion, enhanced warming over the WIO should weaken the zonal SST gradient as well as the Walker circulation due to climatological SST pattern. Why did you say "strengthening the zonal SST gradient" here?

3. Lines 112-113: How did you get this equation for calculating G to define the air-sea interaction? Are you the first to use it? If so, please give more description and evidence of the definition of G equation. If not, please give the citation and also add more explanation.

4. Lines 119-121: Here, G pattern is used to describe the atmospheric damping of SST warming. However, from the equation G pattern only reflects the shortwave damping effect. As we know, the evaporation damping effect due to the high climatological SST in the EIO should also be stronger than WIO. Why don't you examine the damping effect from latent evaporation?

5. Lines 131-132: EIO has deep mean thermocline due to the warm pool there, which can be found in Fig 1e. Why do you say "a shallow mean thermocline" here?

6. Lines 184-185: The effect of ITF in the southern IO has been revealed in Feng et al. (2011) and Dong and McPhaden (2016). Please cite the related papers.

Feng, M., C. Böning, A. Biastoch, E. Behrens, E. Weller, and Y. Masumoto, 2011: The reversal of the multidecadal trends of the equatorial Pacific easterly winds, and the Indonesian Throughflow and Leeuwin Current transports. *Geophys. Res. Lett.*, 38, L11604, doi:10.1029/2011GL047291

Dong, L., and M. J. McPhaden, 2016: Interhemispheric SST gradient trends in the Indian Ocean prior to and during the recent global warming hiatus. *Journal of Climate*, 29(24), 9077-9095, <https://doi.org/10.1175/JCLI-D-16-0130.1>

7. Line 196: To support your statement, please specify the correlation coefficient between the Qsw and SST over the SEIO and its statistical significance.

8. Line 209: Figure 4 occurred at the first time here as Fig. 4d, which is unusual. First occurred in the manuscript should be Fig. 4a. Please re-order the sub-figures of Figure 4.

Reply to three reviewers' comments

We are also grateful to the reviewers for their constructive comments and suggestions to improve the manuscript. We have attempted to address the reviewer's comments that are relevant to the present manuscript by conducting additional analysis (such as, additional numerical experiments using the atmosphere-only model simulations, use of more climate models from CMIP6 archive, comparison of cloud changes with the observations, relative roles of thermal and haline changes in stratification). Here we submit a point-by-point reply to each of the points raised. The original comments are in black *italic* and the replies are marked in blue.

Reviewer #1 (Remarks to the Author):

Manuscript Title: Future Indian Ocean warming patterns

Major comments:

1. This paper points out that the future Indian Ocean warming hotspots are tied to the weakening of the IWC. My major concern is that the conclusions are mostly drawn by concurrent composite maps, which obscure the causal chain and are not convincing enough. The author may consider adding sensitivity experiments using CESM2, including more CMIP models for cross-model validations and/or analyzing lead-lag relationships.

We are tremendously grateful to the reviewer for their comments/suggestions, thoughtful remarks, and thorough evaluation of the manuscript. We have addressed the full set of questions raised by the reviewer in the updated version of the manuscript, and our responses are given below.

First, we have followed the reviewer's suggestion of adding new analyses from a sensitivity experiment to better understand the cause of the warming pattern in the Arabian Sea. In the revised version of the manuscript, we have analyzed atmosphere-only model simulations forced by the prescribed SST changes using AMIP6 simulations (publicly available from

ESGF). The atmosphere-only simulations in AMIP6 are (1) AMIP run with historical observed SST and external forcing from 1979 to 2014, and (2) AMIPP4K simulations that are otherwise the same as the AMIP experiments but SST perturbations consist of a uniform warming of 4°K. We have subtracted AMIP from AMIPP4K to get the response of Indian Ocean Walker Circulation to the uniform SST change (uniform warming) in the simulations. For the AMIP6 uniform SST warming experiments, as the SST uniformly increases over the ocean, it does not affect the mean east-west gradient in the base climate. The AMIP6 models used for the analysis are BCC-CSM2-MR, CNRM-CM6-1, CESM2, CanESM5, E3SM-1-0, HadGEM3-GC31-LL, IPSL-CM6A-LR, TaiESM1, MRI-ESM2-0.

To determine whether the SST warming pattern plays a role in the weakening of Indian Ocean Walker Circulation, or whether the weakening of Indian Ocean Walker Circulation causes the SST warming pattern, the multi-model ensemble mean changes in the uniform SST warming experiment have been analyzed. The uniform SST warming experiment shows the anomalous easterlies over the equatorial central Indian Ocean and south-easterlies over the Arabian Sea (Figure R1a). This pattern of wind changes over the Indian Ocean is analogous to what is seen in CESM2-LE simulations (Main Figure 1). Furthermore, a multi-model ensemble mean longitude-height cross-section of the vertical velocity indicates a weakening of ascending (descending) motion over the eastern (western) Indian Ocean (Figure R1b), representing the slowdown of Indian Ocean Walker circulation. This implies that the weakening of Walker Circulation is unlikely to result only from the warming pattern over the equatorial Indian Ocean. More broadly, as a result of this analysis we can say that the Arabian Sea warming in response to a weakened Indian Ocean Walker Circulation is thereby in large part a forced response to the global mean temperature increase.

As suggested by the reviewer, we also validate our results with the 17 CMIP6 model simulations. The CMIP6 models used for the analysis are BCC-CSM2-MR, CAMS-CSM1-0,

CESM2-FV2, CESM2, CESM2-WACCM, CMCC-ESM2, CanESM5, GISS-E2-1-G, MCM-UA-1-0, MRI-ESM2-0, MPI-ESM1-2-LR, MIROC-ES2L, MIROC6, NorESM2-LM, TaiESM1, UKESM1-0-LL, UKESM1-1-LL. The multi-model ensemble mean indicates similar SST warming patterns in the future climate (Figure 2Ra) to what is found with the CESM2-LE simulations (Main Figure 1a); with a significant warming signal over the Arabian sea (AS) and the Southeastern Indian Ocean (SEIO) and comparatively less warming over the eastern Indian Ocean (EIO). The CMIP6 multi-model ensemble also reveals anomalous easterlies over the central Indian Ocean, representing a weakening of Indian Ocean Walker Circulation. Additionally, we identified for the CMIP6 models an enhanced anomalous northward Ekman transport over the AS under future climate change (Figure R2b). Moreover, we also identified an increase in the cross-equatorial volume transport over the equatorial western Indian Ocean is seen in the 21st century (Figure R2e), which further strengthens the meridional Ekman transport over the AS (Figure R2f). Thus, we have identified that the mechanisms sustaining AS warming are consistent across both the CESM2-LE and CMIP6 model simulations.

Additionally, we have investigated the drivers of forced warming patterns over the SEIO in CMIP6 models. The CMIP6 multi-model mean reveals the enhanced downward shortwave radiative flux and reduced cloud cover over the SEIO. Consistent with what was found with the CESM2-LE, for the CMIP6 models a reduction in cloud cover facilitates an enhanced downward shortwave radiative flux at the surface, and thereby enhances surface warming.

We have also analyzed the lead-lag relationship between the SST over the AS and the Indian Ocean Walker Circulation for the CESM2-LE. The maximum of lead-lag correlation was found to occur at 0-lag with a coefficient of 0.97 (Figure R3), indicating that concurrent Indian Ocean Walker Circulation and SST changes may be significant. While the correlation with detrending data is reduced to 0.89. These results suggest that both the Indian Ocean Walker Circulation

and SST over the AS are positively correlated and the relationship could be intensified by the global warming effect.

Figure R1 Circulation changes in the uniform warming atmosphere-only model simulations. Multi-model ensemble mean changes in (a) surface temperature (shaded, °C) overlain by surface wind-stress (vectors, N/m²) for AMIP6 simulations forced by prescribed uniform SST warming. (b) The multi-model ensemble mean longitude-height cross-section (5°N~5°S) of vertical velocity (Pa/s, shaded) and zonal-vertical velocity (vectors) from 1000-100 hPa in AMIP6 simulations. The magnitude of vertical velocity is lesser than zonal velocity, so, vertical velocity is multiplied by the factor 800. Positive (negative) values represent the anomalous descending (ascending) motion. (c) The meridional average of the multi-model ensemble mean of surface wind-stress over the equatorial region (5°N~5°S). Positive (negative) values represents the anomalous westerlies (easterlies). The dark (light) shading across the line represents the 0.5 (1) inter-ensemble standard deviation.

Figure R2: Drivers of warming patterns in the Indian Ocean in CMIP6 model simulations. Ensemble mean changes in (a) SST (shaded) and surface wind stress (vectors) (b) wind stress curl (shaded) and Ekman transport (vectors) (c) short wave radiative flux and (d) total cloud cover for CMIP6 (13 models). The ensemble mean changes are calculated by taking the difference between the future (2080-2100) and historical (1980-2000) periods for all the simulations. Stippling in (a-d) indicates the regions where the difference between future and historical periods is significant at the 95% confidence level based on a two-sample *t*-test on the time series of ensemble means. The bottom panels show the ensemble mean area averaged time series of (f) cross-equatorial volume transport over the equatorial WIO, and (e) meridional and zonal Ekman transport over the Arabian sea. The dark shading across the time series represents the one ensemble standard deviation.

Figure R3: Lead-lag relationship of SST over the AS and Indian Ocean Walker Circulation (IOWC). Time lagged correlation coefficients between annual-mean area-averaged SST over the AS and Indian Ocean Walker Circulation Index time series over 1850-2100 for the 50 ensemble members of the CESM2 LE. Both the time-series are relative to an 1850-2014 baseline, and a 3-year running mean is applied to the time series. For SST time-series mean values over the Indian Ocean are subtracted from the corresponding area averaged values over the AS. The Indian Ocean Walker circulation Index is calculated by taking the difference between the area-averaged meridional velocity at 500 hPa in the eastern Indian Ocean (94°E-104°E, 5°S-5°N) and the western Indian Ocean (40°E-50°E, 5°S-5°N).

2. Calculating the relative roles of thermal and haline changes on stratification is inappropriate because T and S are not mutually independent. There are other ways to compare contributions from the two, such as the Monin-Obukhov ($M-O$) stability parameter (the ratio of thermocline depth to the Monin-Obukhov length).

The reviewer's comment is much appreciated. Here, we intended to compare relative roles of changes in temperature stratification and salinity stratification in changes in the density stratification which in turn regulates the intensity of the vertical mixing. In the calculation, we assumed the linear relationship between temperature and salinity to density (Equation 8). This

relationship is well held under the finite changes in temperature and salinity and is well established and used in previous studies (Yamaguchi and Suga, 2019) in this context. Moreover, in our case, the sum of thermal and haline contributions corresponds well to the density stratification changes itself (Supplementary Fig. 11), which means the linear assumption is held here too.

To our pour knowledge about the Monin-Obukhov length and stability parameter, it is used to evaluate the relative contribution of buoyancy forcing and mechanical forcing to upper-ocean mixing (i.e., diagnosing observational surface mixed layer). It seems that it does not fit to our intention (decomposition of changes in density stratification into thermal and haline contributions) here. We apologize if we didn't make ourselves clear about the reviewer's comment, but we would appreciate if you could explain more or give any references about it.

As per the reviewer's suggestion, we have also evaluated the relative contributions of thermal and haline/freshwater flux changes to stratification changes using the Monin-Obukhov stability parameter. The Monin-Obukhov length (M-O length, L) is a measure of the relative importance of wind forcing to buoyancy forcing (Sutherland et al., 2013, Shee et al., 2018). It is negative (positive) for stabilizing (destabilizing) conditions. The following equations are used for the calculation of M-O stability parameter and the definitions of the symbols are given in the accompanying box.

$$B_o = B_T + B_S$$

$$B_o = \frac{\alpha Q_{net}}{C_p} + \rho\beta(P - E)S$$

$$U^* = \sqrt{\frac{\tau}{\rho}}$$

$$\mathcal{L} = - \left(\frac{(U^*)^3}{\kappa g B_o} \right)$$

- B_o = Surface Buoyancy Flux
- B_T = Surface Thermal Flux
- B_S = Surface Haline Flux
- α = Thermal expansion coefficient
- β = Haline Contraction coefficient
- ρ = Surface density
- Q_{net} = Net surface heat flux
- P = Precipitation
- E = Evaporation
- S = Surface salinity
- C_p = Specific heat of water (3990 J/kg/C)
- U^* = Frictional velocity
- τ = Surface wind stress
- κ = von Karman's constant (0.41)
- g = acceleration due to gravity (9.8 m/s²)
- \mathcal{L} = Monin-Obukhov Length

Our analysis (Figure R4a) indicates that the total surface buoyancy flux into the AS increases under a warming climate. A decomposition of surface buoyancy flux reveals that the haline/freshwater flux component (mainly associated with precipitation) provides the dominant contribution to the buoyancy flux changes (Figure R4c), with an only second-order contribution from the thermal flux component (Figure R4b). The thermal flux changes are identified using the net surface heat flux, which decreases over the AS in the warming climate (Main Figure 2a(i)). The haline/freshwater flux is calculated using precipitation minus evaporation (P-E), which exhibits an increase over the AS in the warming climate (Supplementary Figure 10), therefore these results are not surprising. Moreover, the surface wind stress decreases over the AS in the warming climate (Figure 1a), with the change being clearly reflected in the frictional velocity as well (Figure R4d). The calculation of M-O length using both the total buoyancy and the frictional velocity yields negative values, with this corresponding to stabilizing conditions (Figure R4e). Finally, the M-O length (L) is compared with MLD (h), and their ratio (L/h) is defined as a stability parameter. Values of $L/h > 1$ indicate that the mixed layer depth is dominated by wind mixing, whereas values of $L/h < 1$ indicate that the mixed layer changes are dominated by buoyancy forcing. Our results show that for conditions in both the historical and future periods ($L/h < 1$), the mixed layer depth is determined largely by the buoyancy forcing, and in particular by the freshwater forcing associated with changes in precipitation over the region (Figure R4f).

Importantly, our ocean heat budget analysis implicates a role for ocean mixing in the anomalous surface warming in the AS under climate change. The AS region corresponds to the location of the upwelling branch of the cross-equatorial overturning cell, which weakens under future climate change. Thus, our identification of a dominant role for freshwater forcing

relative to heating for the surface complements the view for the interior (described in the submitted manuscript) whereby there was more parity between heat and salinity in setting the interior stratification.

- Shee, A., Sil, S., Gangopadhyay, A., Gawarkiewicz, G., & Ravichandran, M. (2019). Seasonal evolution of oceanic upper layer processes in the northern Bay of Bengal following a single Argofloat. *Geophysical Research Letters*, 46, 5369–5377. <https://doi.org/10.1029/2019GL082078>
- Sutherland, G., Ward, B., & Christensen, K. H. (2013). Wave-turbulence scaling in the ocean mixed layer. *Ocean Science*, 9(4), 597–608. <https://doi.org/10.5194/os-9-597-2013>

Figure R4 Forced response in view of the contributions of thermal and haline/freshwater fluxes to surface stratification. Individual markers in the scatter plots, representing the individual 50 ensemble member of CESM2-LE, indicate the temporal means of the historical and future periods for (a) total buoyancy flux, (b) thermal component of surface buoyancy flux, (c) haline/freshwater component of surface buoyancy flux, (d) frictional velocity, (e) Monin-Obukhov length and (f) ratio of Monin-Obukhov length to the mixed layer depth. Red solid dots denote the ensemble mean.

Minor comments:

1. Line 81: “strengthening and zonal SST gradient” —> “weakening the zonal SST gradient”?

Thank you for pointing this out. It has now been corrected in the revised version.

2. Line 96: how well can the CESM2 simulate the historical IO warming patterns?

The reviewer's question here is greatly appreciated. To investigate historical warming patterns, we calculated empirical orthogonal functions (EOFs) of SST during the satellite period 1982-2020 using both observational products and the CESM2-LE over the Indian Ocean. The CESM2-LE represents well the general characteristics of historical warming patterns in the Indian Ocean (Figure R5). The increased warming signal in the AS and equatorial WIO is very well simulated in the simulations. The simulations also capture the decrease of SST over the eastern equatorial Indian ocean (0~10°S). Furthermore, the warming signal over the southern area (10°S~25°S, 55°E~85°E) is well captured in the CESM2-LE simulations.

Figure R5 Historical warming pattern in the Indian Ocean. Empirical orthogonal function (EOF) of annual mean SST of (a) HadISST and (b) the ensemble mean of 50 members of CESM2-LE over the period 1982 to 2020.

3. Line 125: No corresponding Figures for “the enhanced damping...”. Fig.1 only shows the climatological G ? Suggest showing its future changes as well.

Thank you for the suggestion. In the revised version, we have shown the future changes in the shortwave damping over the EIO. All of the members (and thereby the ensemble mean) indicate an enhanced atmospheric damping of SST anomalies under future climate change. Additionally, we examine the relationship between future changes in G and SST over the EIO (see Fig. R6), revealing a significant negative correlation. That suggests that negative feedback of atmospheric damping on SST anomalies will be enhanced under future climate change.

Figure R6 Future changes in the pattern for *G* and its relationship with SST over EIO. **(a)** The bar plot shows the future changes (future minus historical) in *G* over the EIO for 50 ensemble members and the ensemble mean. The negative values represent the strong damping of SST anomalies. The error bar shows the ensemble standard deviation. **(b)** Scatter plot shows the relationship between the *G* and SST changes over the EIO. Each dot in the scatter plot represents an individual ensemble member and the line represents the slope across the ensemble members.

4. Line 343: Have the author looked at other CMIP6 models?

As per the reviewer's suggestion, we conducted an additional analysis to compare the results of abrupt-4xCO₂ and pre-industrial Control runs for CESM2 with other CMIP6 models. The results from the collection of CMIP6 models (not including CESM2) are comparable to the CESM2, demonstrating the robustness of our findings across models. An analysis of the CMIP6

multi-model mean reveals a weakening of meridional pressure gradient (Figure R7a), a reduction in the cross-equatorial volume transport (Figure R7b), and an enhanced anomalous northward Ekman transport (Figure R7c), all of which is consistent with the results found with CESM2, and thereby with the CESM2-LE.

Figure R7 Temporal evolution of pertinent variables showing the difference between CMIP6 abrupt-4xCO₂ and piControl experiments. Temporal evolution of the difference between abrupt-4xCO₂ and pre-industrial Control simulation of (a) meridional pressure gradient (difference between southern Indian Ocean (10S-35S, 40E-100E) and northern Indian Ocean (10N-30N, 40E-100E)), (b) cross-equatorial volume transport over equatorial WIO, and (c) meridional Ekman transport over the AS. An 11-year running mean is applied to each time series, and the shading indicates one standard deviation. Less negative values in (b and c) indicate a weakening of southward transport.

5. Line 375: since the heat budget analyses are applied to the upper 65m below the mixed layer depth, it is necessary to check that the ocean heat content variation within the upper 65m is in high consistency with the sea surface warming patterns.

Thank you for this suggestion. A plot describing the ocean heat content changes is now added in the revised version (Figure R8). We found that the pattern of ocean heat content changes within the upper 65m depth and SST changes are highly consistent, confirming the reviewer's point.

Figure R8 Climatology and changes of ocean heat content in the upper 65m in CESM2-LE simulations. (a) Ensemble mean climatology of ocean heat content for the 50 members of CESM2-LE during the historical period (1980-2000). (b) Ensemble mean changes of ocean heat content. The changes are calculated by taking the difference between the future (2080-2100) and historical (1980-2000) periods. Ocean heat content is calculated as: $OHC = c_p \int_{-65}^0 \rho(z)T(z)dz$, where c_p is the specific heat capacity of water, $\rho(z)$ is the sea water density profile and $T(z)$ is the temperature profile.

6. Figure 1: how about the ensemble spread?

Thank you for the suggestion. Ensemble spread is now added for the SST changes.

7. Figure 2: better to give the time series of Temp because it is hard to see the Temp change with time by the tendency.

Thank you for the suggestion. A time-series plot of vertically averaged (0~65m) temperature over the AS and SEIO (Figure. R9) is now added to the sub-section of main figure 1.

Figure R9 Time-series of vertically and area-averaged temperature over the different regions in the Indian Ocean. Time series of vertically and area-averaged (0~65m) temperature over the AS (red) and SEIO (black) during the period 1850 to 2100 for the mean of 50 ensemble members. Note that the IO mean has been removed from each vertically area-averaged time series over the AS and SEIO. Both of the timeseries are calculated relative to a 1850-2014 baseline, and subsequently a 15-years running has been applied. The shading across the time-series represents the one standard deviation level.

8. Figure 5: something wrong with the caption : “G pattern (shaded)” 、 “SST (°C, shaded)”. Consider using shadings for SSTs, and contour lines for G.

Thank you for pointing out this mistake. It has now been corrected in the revised version of manuscript. Also, we have changed the schematic diagram considering shadings for SSTs and contour lines for G.

9. Lines 101-102 “the projected intensified surface warming is confined to the northern

WIO”, I think such a description is not proper. As seen in Figure 1, there is also intensified surface warming in equatorial WIO. Thinking about rephrasing the description.

Thank you for the careful observation. Line 101-102 *“It warrants mentioning that the projected intensified surface warming is confined to the northern WIO, i.e., the AS, and not to the equatorial WIO, as was found in previous studies.”* is now replaced with *“The projected surface warming is larger in the AS, as compared to the equatorial WIO, as has been pointed out in previous studies”*

10. Lines 208-209, You mention that the reduction in the low-cloud cover is caused by the low-level anomalous anticyclonic circulation over the SEIO. However, I see anomalous cyclonic circulation over the SEIO.

We thank the reviewer for this comment. The *“anomalous anticyclonic circulation”* has now been replaced with *“anomalous cyclonic circulation”* in the revised version.

11. Lines 351-352, please double-check the expression and sign of the first and second term on the right-hand side in equation (1).

Thank you. We have double-checked the sign of first and second terms in equation (1) and can confirm that they are correct.

12. Line 393-395, why is the cross-equatorial volume transport in the WIO calculated on the 5° transect? Does 5° mean 5°N?

Thank you for your comment. As is well-known, the relatively low-pressure signature of the Northern Hemisphere associated with the enhanced thermal response of the subcontinental land mass and the relatively high pressure over the Southern Hemisphere (Mascarenes high) creates a strong pressure gradient between the two hemispheres which further sustains a cross-

equatorial winds over the equatorial western Indian Ocean. It has been previously reported in numerous studies (as was mentioned in the submitted version of the manuscript) that the degree of intensity of anticyclonic circulation around the high-pressure system in the Southern Hemisphere can determine whether the cross-equatorial/meridional heat transport in the western Indian Ocean weakens or strengthens. Here in our analysis, the 5° transects refer to the difference of meridional gradient of zonal wind stress along 5°N and 5°S over the equatorial western Indian Ocean (50°E - 75°E). The increase of forced cross-equatorial volume transport or weakening of southward transport over the equatorial domain is relatively insensitive to whether one chooses 6°N and 6°S or 4°N and 4°S instead. Following the suggestion of the reviewer, we have now provided a clarification of our description of the 5° transects in the revised text.

13. Line 588, It is better to tell people how you define mixed layer depth.

Thank you for your suggestion. Here, we use the thickness of the turbulent mixing layer calculated by the ocean mixed layer model (KPP scheme) in the CESM2 as the definition of the mixed layer depth.

14. Line 152 in the supplementary, “the slope across the across the ensemble members”, double “across the”.

Thank you for pointing out this. This has now been corrected in the revised version.

Reviewer #2

This study investigates the formation mechanism of the Indian Ocean warming pattern under global warming by analyzing CESM2 large ensemble simulations. The authors conclude that the weak equatorial Indian Ocean warming is due to negative cloud-SST feedback, the enhanced southeast Indian Ocean warming is caused by reduced cloudiness, and the strong Arabian Sea warming is attributed to changes in meridional advection and vertical mixing. The Indian Ocean warming pattern formation has been analyzed extensively in the past, and I am afraid that I have not found significant enough contribution from this study that warrants publication in Nature Communications. Besides, I have several concerns about the conclusions, which are related to model biases in simulating clouds and the atmosphere-ocean coupling processes. Based on my evaluation, I don't find this manuscript suitable for publication in its current form. I have listed my comments below.

We appreciate the reviewer's thorough evaluation of the manuscript. We believe that the concerns of the reviewer have been addressed in the updated version of the manuscript and in our response below.

Specific comments:

1. Slowdown of the Indian Ocean Walker cell is a robust feature that can be found in most climate model projections, and as suggested by the authors, it plays an important role in affecting the Indian Ocean warming pattern. This is not too surprising and has been discussed in previous literatures already. However, I think the important question that needs addressed is what causes changes in the Indian Ocean winds. It seems that the authors have attributed the easterly wind anomalies to the weaker EIO warming (L36-40), but since atmosphere and ocean are fully coupled, the wind changes could in turn induce the weaker EIO warming in the first place. In particular, the well-known El Niño-like warming in the tropical Pacific could

induce such Indian Ocean wind changes and subsequently affect Indian Ocean warming pattern. Hence, I don't think the analysis in this study could fully support the conclusion that the weaker EIO warming is due to negative cloud-SST feedback.

Thank you for the opportunity to respond to the reviewer's comments. To address the reviewer's concerns as to whether an El-Niño like warming in the tropical Pacific can induce Indian ocean wind changes or weaker EIO warming, we conducted further analyses of suitable atmosphere-only model simulations to identify the cause-and-effect relationship. In the revised version of the manuscript, we have now included our analysis of the atmosphere-only model simulations forced by prescribed SST changes using AMIP6 simulations available through ESGF. The atmosphere-only simulations from AMIP6 are (1) AMIP run that uses historical observed SST and external forcing from 1979 to 2014; (2) AMIPP4K simulations where the SST boundary forcing is subject to a uniform warming of 4°K; (3) AMIPFuture4K simulations that are otherwise the same as the AMIP experiments for the historical period, except that SSTs are subject to a composite of SST warming pattern derived from coupled models, scaled to an ice-free ocean mean of 4K. We have subtracted AMIP from AMIPP4K to get the response to uniform SST perturbations (uniform warming) in AMIP6 simulations. Also, to get the response of pattern SST change (pattern warming), we subtract AMIPP4K from AMIPFuture4K simulations. In the uniform warming experiment, the SST uniformly increases over the ocean; it does not affect the mean east-west gradient in the base climate. However, pattern warming experiment represents El-Niño like warming conditions, similar to coupled model simulations. Thus, these experiments provide an invaluable tool with which to identify whether the weaker EIO warming response is modulated by greenhouse gas (GHG)-induced warming or El Niño-like warming. The AMIP6 models used for the analysis are BCC-CSM2-MR, CNRM-CM6-1, CESM2, CanESM5, E3SM-1-0, HadGEM3-GC31-LL, IPSL-CM6A-LR, TaiESM1, MRI-ESM2-0.

In addition, to validate the robustness of our results, we also investigated the real-time decadal climate prediction project (dcpp) simulations to understand longer timescale variations and their regional imprints. The dcpp project includes idealized model experiments to explore the mechanisms determining global and regional climate responses to modes of decadal variability in the Pacific and Atlantic Oceans. Here, we used control (dcppC-pac-control) and sensitivity (dcppC-ipv-pos) experiments to investigate the role of decadal variability of the eastern Pacific in modulating regional climate variations, in particular over the Indian Ocean.

Our analysis reveals that both the uniform and pattern warming experiments exhibit anomalous surface easterly winds over the equatorial central Indian Ocean. On the other hand, south-easterly winds over the Arabian Sea are only found in the uniform warming experiment (Figure R1a, b), with this pattern of change being similar to what is found for coupled model simulations. Moreover, both sets of experiments represent a weakening of the Walker circulation, with the magnitude of anomalous descending motion over the EIO is greater with uniform warming than with pattern warming, due to the difference in the tropical-mean or Indian Ocean-mean warming (Figure R1d). This demonstrates that both GHG-induced warming and an El Niño-like warming pattern contribute to weakening the ascending motion over the EIO, with the former exerting somewhat greater influence. We also wish to point out that dcpp simulations don't show a distinct cooling over the EIO (Figure R1c), which is seemingly inconsistent with the hypothesis that El Niño-like warming-induced anomalous zonal wind over the EIO drives a cooling there (or weaker EIO warming in the presence of GHG increase). This discrepancy might indicate that El Niño-like warming is not a main factor leading to weaker EIO warming, and that a negative cloud-SST feedback mechanism might play a greater role in the weaker EIO warming. To elucidate this further, we investigated the relationship between SST over the equatorial eastern Pacific and surface wind over the central Indian Ocean in the CESM2-LE simulations (Figure R1e). The results indicate that the

relationship between SST over the equatorial eastern Pacific and surface wind over the central Indian Ocean becomes weaker under future climate change. This also demonstrates that weaker EIO warming is not primarily driven by El Niño-like warming in the future climate.

Figure R1 Indian Ocean wind changes and their linkage to the Pacific Ocean in coupled model simulations and atmosphere-only model experiments. Multi-model ensemble mean changes in surface temperature (shaded, °C) overlain with surface wind-stress (vectors, N/m^2) in (a) AMIP6 uniform SST warming experiment (AMIPP4K minus AMIP), (b) the AMIP6 pattern SST warming experiment (AMIPFuture4K minus AMIPP4K), (c) the decadal Pacific variability changes (dcppC-ipv-pos minus dcppC-pac-Control) simulations to determine the idealized impact of positive interdecadal Pacific variability. (d) Multi-model ensemble mean changes in meridional average ($5^{\circ}N\sim 5^{\circ}S$) of mid-tropospheric vertical velocity at 500 hPa in the uniform (red) and pattern (black) SST warming experiments. Positive (negative) values represent an anomalous descending (ascending) motion. The shading across the line represents the one standard deviation level. The relationship between the area-averaged SST over the eastern equatorial Pacific Ocean ($5^{\circ}N\sim 5^{\circ}S$, $190^{\circ}E\sim 270^{\circ}E$) and surface zonal wind stress over the central equatorial Indian Ocean ($5^{\circ}N\sim 5^{\circ}S$, $60^{\circ}E\sim 90^{\circ}E$) during the (e) historical (1980-2000) and (f) future (2080-2100) period, separately. The red dots represent each ensemble member and the black line indicates the linear regression. Correlation coefficients and corresponding *p*-values during the both historical and future period are also indicated.

2. It seems that cloud changes play an important role in affecting SST changes. However, current generation of climate models still have quite large uncertainties in simulating cloud changes, especially low clouds. How this may affect the conclusions in this study, especially for the southeast Indian Ocean where the reduced low clouds seem crucial in causing SST warming?

The reviewer's feedback is greatly appreciated. We agree with the reviewer that current generation of climate models have significant uncertainties in representing cloud changes. Among the CMIP6 generation of Earth system models, CESM2 has been comprehensively identified with objective skill metrics to have relatively very strong fidelity to observational constraints (Fasullo *et al.*, 2020), and is thereby deemed to be an appropriate choice for our process-based study. Furthermore, recent studies have also highlighted the skill of CESM2 in representing the cloud radiative effect, with this serving as a critical benchmark for cloud

feedbacks (*Gettleman et al., 2019*). Also, CESM2 produces an excellent representation of the current climate system (*Danabasoglu et al., 2020*; also Fig. 1 of *Rodgers et al., 2022*).

Here, we also compared the preset-day time-averaged vertically integrated low and total cloud cover patterns in the CESM2-LE simulations with the observationally-anchored ERA-5 product over the time-period 1980 to 2020. CESM2-LE represents a reduction in the low cloud over most of the tropical region, that is very similar to what is found in observations (Figure R2a-b). Moreover, the large-scale pattern of total cloud cover is well represented in CESM2-LE relative to the observationally-based products (Figure R2b-d). Furthermore, observations lie within the ensemble spread for zonally and meridionally integrated low and total cloud cover (Figure R2e-h). This shows that although model-observation differences exist in cloud climatology and variability, CESM2 performance is better than other models participating in the CMIP6.

- *Fasullo, J. T.: Evaluating simulated climate patterns from the CMIP archives using satellite and reanalysis datasets using the Climate Model Assessment Tool (CMATv1), Geosci. Model Dev., 13, 3627–3642, <https://doi.org/10.5194/gmd-13-3627-2020>, 2020.*
- *Gettelman, A., Hannay, C., Bacmeister, J. T., Neale, R. B., Pendergrass, A. G., Danabasoglu, G., et al. (2019). High climate sensitivity in the Community Earth System Model Version 2 (CESM2). Geophysical Research Letters, 46, 8329–8337. <https://doi.org/10.1029/2019GL083978>*
- *Danabasoglu, G., Lamarque, J.-F., Bacmeister, J., Bailey, D. A., DuVivier, A. K., Edwards, J., et al. (2020). The Community Earth System Model Version 2 (CESM2). Journal of Advances in Modeling Earth Systems, 12, e2019MS001916. <https://doi.org/10.1029/2019MS001916>*
- *Rodgers, K. B., Lee, S.-S., Rosenbloom, N., Timmermann, A., Danabasoglu, G., Deser, C., Edwards, J., Kim, J.-E., Simpson, I. R., Stein, K., Stuecker, M. F., Yamaguchi, R., Bóday, T., Chung, E.-S., Huang, L., Kim, W. M., Lamarque, J.-F., Lombardozzi, D. L., Wieder, W. R., and Yeager, S. G.: Ubiquity of human-induced changes in climate variability, Earth Syst. Dynam., 12, 1393–1411, <https://doi.org/10.5194/esd-12-1393-2021>, 2021*

Figure R2 Comparison of vertically integrated low and total cloud cover between ERA-5 and CESM2-LE simulations. (a) Climatology of vertically integrated low-cloud cover (fraction) during the time period 1980-2020 in the ERA-5 reanalysis. (b) Ensemble mean climatology of vertically integrated low-cloud cover (fraction) during the time period 1980-2020 for the 50 ensemble members. Note that in both (a) and (b) 60°N-60°S mean values are removed from the climatological values. (c, d) are same as (a, b) but for vertically integrated total cloud cover. (e) Zonally integrated (0E-360°E) low-cloud cover in ERA-5 (red) and ensemble mean in

CESM2-LE (black). (f) same as (e) but for the total-cloud cover. (g) Meridionally integrated (60°N~60°S) low-cloud cover in ERA-5 (red) and ensemble mean in CESM2-LE (black). The inset plot shows the meridionally integrated low-cloud cover in the southern hemisphere (equator~35°S) only. The region of southern Indian Ocean (40°E~110°E) is highlighted with light red shading. (h) same as (g) but for the total-cloud cover. The dark (light) grey shading across the line in (e, f, g and h) represents the 0.5 (1) inter ensemble standard deviation.

3. The wind changes seem important for the AS warming, which have been further attributed to the Arabian land warming (L138-139). I find this argument not quite convincing. For instance, the land warming over India is also very strong, yet we don't see winds converging toward the region. It seems to me that the easterly wind anomalies over the AS could be driven by the strong SST warming instead. I think more evidence is needed to support the argument.

Thank you for your further valuable comments on our manuscript. Under future climate change, the ensemble mean changes of the surface temperature are two to three times stronger over the Arabian Peninsula than over the Indian domain (Figure R3a). Additionally, the ensemble mean changes in sea-level pressure indicate a strong anomalous low-pressure signal that is centered over the Arabian Peninsula region, accompanied by anomalous high-pressure over the Indian domain. Both surface temperature and sea-level pressure patterns mirror each other over the Arabian Peninsula. Previous studies have identified similar projected changes in the surface temperature and sea-level pressure over the Arabian Peninsula in the other model simulations (*Sharma et al., 2022*). The larger warming over the land is attributed to a larger lapse rate over the land rather than a lower heat capacity of the land (*Joshi et. al., 2008*). The higher temperature over land reflects the fact that the troposphere over land is often drier than that over the ocean, and a higher lapse rate results in more surface warming, especially in dry areas, such as the Arabian Peninsula. The strong anomalous land warming and anomalous low-pressure over the Arabian Peninsula cause the anomalous easterlies over the central Indian

Ocean to turn northwestward over the AS. Moreover, the uniform SST warming experiment also shows the south-easterlies over the AS are very similar to those in the coupled model simulations (Figure R1a). This further implies that the that the surface wind changes over the AS are mainly due to GHG-induced warming, and unlikely to result from the strong SST warming.

- *Sharma, S., Ha, KJ., Cai, W. et al. Local meridional circulation changes contribute to a projected slowdown of the Indian Ocean Walker circulation. npj Clim Atmos Sci 5, 15 (2022).*
- *Joshi, M. M., Gregory, J. M., Webb, M. J., Sexton, D. M. H. & Johns, T. C. Mechanisms for the land/sea warming contrast exhibited by simulations of climate change. Clim. Dyn. 30, 455–465 (2008).*

Figure R3 *Surface temperature, wind and sea level pressure changes in the CESM2-LE simulations. Ensemble mean changes of (a) surface temperature (shaded) and surface wind (vectors) (b) sea level pressure (shaded) and surface wind (vectors) for the 50 ensemble members of CESM2-LE. The mean state changes are calculated by taking the ensemble mean difference between the future (2080-2100) and the historical (1980-2000) periods. The color vectors represent the magnitude of the surface wind.*

4. Heat budget analysis is crucial for drawing conclusions in this study. However, I notice that the budget is performed with a fixed mixed layer depth (L161-162), which is much deeper than the actual MLD (Fig. 1). I wonder how the results may change if the authors use the MLD in the heat budget instead.

Thank you for your comment. We used the fixed depth to avoid un-closure of the heat budget due to the use of the time-varying the lower bound of the domain (i.e., mixed layer depth) in the mixed layer heat budget. Indeed, terms in the right-hand side of the fixed depth heat budget and mixed layer heat budget differ each other (e.g., entrainment due to mixed layer deepen). We conducted the fixed depth heat budget because of limited saved output from CESM2-LE which is saved mostly monthly and is not enough to calculate full terms of mixed layer heat budget without occurring huge errors. We think our conclusions will be unchanged even if the MLD heat budget is used, based on the following two points. 1) Under future climate change, the mixed layer depth over the Arabian Sea remains relatively unchanged. 2) We found that the pattern of ocean heat content changes within the upper 65m depth, and SST changes are quite consistent (Figure R4 and Main figure 1a). The new analyses discussed here confirm the utility of the calculation done over the upper 65m.

Figure R4 Climatology and changes of ocean heat content in the upper 65m in CESM2-LE simulations. (a) Ensemble mean climatology of ocean heat content for the 50 members of CESM2-LE during the historical period (1980-2000). (b) Ensemble mean changes of ocean heat content. The changes are calculated by taking the difference between the future (2080-2100) and historical (1980-2000) period. Ocean heat content is calculated as: $OHC = c_p \int_{-65}^0 \rho(z)T(z)dz$, where c_p is the specific heat capacity of water, $\rho(z)$ is the sea water density profile and $T(z)$ is the temperature profile.

Reviewer #3 (Remarks to the Author):

Review of the manuscript “Future Indian Ocean warming patterns” by Sharma et al.

Reviewer comments:

This paper examines the mechanism for the non-uniform warming pattern over Indian Ocean. Enhanced warming hotspots are found in the Arabian Sea (AS) and the southeastern Indian Ocean (SEIO). The heat budget analysis is conducted to identify the relative contribution of atmospheric and oceanic processes. The ocean heat transport mainly contributes to the enhanced AS warming, while the cloud-shortwave radiation plays a key role in the SEIO warming. The manuscript is well written, and the finding is interesting and robust. However, some processes still remain unclear and more evidences are needed. I recommend a major revision before the paper can be published in Nature Communications.

The reviewer's comments/suggestions, thoughtful remarks, and thorough evaluation of the manuscript are greatly appreciated. We believe that the updated version of the manuscript and our response below fully address all of the reviewer's comments.

Major comment:

1. The main concern is about the mechanism for the enhanced warming over AS. In this mechanism, the weakened IO Walker circulation is the reason for anomalous heat transport responsible for the AS warming. The question is whether the Walker circulation is induced by the enhanced warming over the WIO or the AS? If so, they are coupled and we cannot say which is the cause. Maybe an experiment with uniform SST warming can answer this question. I suggest to adding an analysis based on a uniform SST warming experiment. If the weakened IO Walker circulation still exists in the uniform SST warming experiment, we can believe the weakened IO Walker circulation is the cause of the enhanced AS warming, and your argument would make sense.

The reviewer's suggestion is greatly appreciated. We have addressed this concern by adding a sensitivity experiment to better understand the cause-and-effect relationship. In the revised version of the manuscript, we examined the AMIP6 atmosphere-only model simulations forced with prescribed SST changes. The atmosphere-only simulations chosen from AMIP6 are (1) the AMIP run with historical observed SST and external forcing from 1979 to 2014, and (2) the AMIPP4K simulations involving AMIP experiments where SSTs are subject to uniform warming of 4°K. We subtract AMIP from AMIPP4K to get the response of Indian Ocean Walker Circulation to a uniform SST change (uniform warming). The precise point of interest is that for uniform SST warming experiments, the mean east-west gradient in SST for the base climate state is not impacted. The AMIP6 models used for the analysis are BCC-CSM2-MR, CNRM-CM6-1, CESM2, CanESM5, E3SM-1-0, HadGEM3-GC31-LL, IPSL-CM6A-LR, TaiESM1, MRI-ESM2-0.

To determine whether changes in the zonal gradient of SST under global warming plays a role in the weakening of Indian Ocean Walker Circulation, or whether the weakening of Indian Ocean Walker Circulation causes the SST warming pattern, the multi-model ensemble mean changes in the uniform warming experiment are analyzed. The uniform SST warming perturbation experiment reveals the presence of anomalous easterlies over the equatorial central Indian Ocean and south-easterlies over the Arabian Sea (Figure R1a). Overall, the pattern of wind changes over the Indian Ocean is analogous to what is seen in CESM2-LE (Main Figure 1). Furthermore, a longitude-height cross-section of the vertical velocity indicates a weakening of ascending (descending) motion over the eastern (western) Indian Ocean (Figure R1b), representing the slowdown of Indian Ocean Walker circulation. This implies that weakening of the Walker Circulation can be expected as a response to uniform SST warming. Stated differently, a weakened Indian Ocean Walker Circulation is one of the primary causes of enhanced Arabian Sea warming under future climate change.

Figure R1 Circulation changes in the uniform warming atmosphere-only model simulations. Multi-model ensemble mean changes in (a) surface temperature (shaded, $^{\circ}\text{C}$) overlain by surface wind-stress (vectors, N/m^2) in AMIP6 simulations forced by prescribed uniform SST warming. (b) The multi-model ensemble mean longitude-height cross-section (5°N - 5°S) of vertical velocity (Pa/s , shaded) and zonal-vertical velocity (vectors) from 1000-100 hPa in AMIP6 simulations. Vertical velocity is multiplied by the factor 800. Positive (negative) values represent the anomalous descending (ascending) motion. (c) The multi-model ensemble mean meridional average of surface wind-stress over the equatorial band bounded by 5°N - 5°S . Positive (negative) values represents anomalous westerlies (easterlies). The dark (light) shading across the line represents the 0.5 (1) inter-ensemble standard deviation.

2. Line 81: In my opinion, enhanced warming over the WIO should weaken the zonal SST gradient as well as the Walker circulation due to climatological SST pattern. Why did you say “strengthening the zonal SST gradient” here?

Thank you for pointing out this. It is now corrected in the revised version of the manuscript.

3. Lines 112-113: How did you get this equation for calculating G to define the air-sea interaction? Are you the first to use it? If so, please give more description and evidence of the definition of G equation. If not, please give the citation and also add more explanation.

The reviewer's question is greatly appreciated. In the revised version we have provided an expanded explanation of G and cited the relevant previous studies as well. Some of the components of the calculations we consider with G have been applied in previous study by other authors, but our specific equation and formal definition of G is new. This information is now clarified in the revised text of the manuscript.

The text now states: “*we estimate the present-day air-sea interaction strength ($G = \langle \dot{Q}_{sw} \times S\dot{S}T \rangle / \langle \dot{Q}_{sw}^2 \rangle$ where \dot{Q}_{sw} and $S\dot{S}T$ represent the shortwave radiative flux and SST perturbation from the climatological mean and the bracket $\langle \dots \rangle$ refers to the ensemble mean. Heat flux anomalies in the Indian Ocean usually serve as negative feedback for SST anomalies (Lloyd et al., 2009, Bayr et al., 2021). . For instance, a positive SST anomaly in this area causes an enhancement of clouds, which in turn reduces shortwave fluxes, and in turn a surface ocean cooling tendency”*

- *Lloyd J, Guilyardi E, Weller H, Slingo J (2009) The role of atmosphere feedbacks during ENSO in the CMIP3 models. Atmos Sci Lett 10(3):170–176. <https://doi.org/10.1175/JCLI-D-11-00178.1>*
- *Bayr, T., Drews, A., Latif, M. et al. The interplay of thermodynamics and ocean dynamics during ENSO growth phase. Clim Dyn 56, 1681–1697 (2021). <https://doi.org/10.1007/s00382-020-05552-4>*

4. Lines 119-121: Here, G pattern is used to describe the atmospheric damping of SST warming. However, from the equation G pattern only reflects the shortwave damping effect. As we know, the evaporation damping effect due to the high climatological SST in the EIO should also be stronger than WIO. Why don't you examine the damping effect from latent evaporation?

Thank you for your suggestion. We have decomposed the latent heat flux (Q_{lh}) into the atmospheric part ($Q_{lh}^w = \langle \overline{Q_{lh}} \times \hat{W} \rangle / \langle \overline{W} \rangle$, where $\overline{Q_{lh}}$ and \overline{W} are the climatological latent heat flux and the surface wind speed and \hat{W} is the perturbation of surface wind speed from the mean) and the oceanic part ($Q_{lh}^o = \langle \alpha \overline{Q_{lh}} \hat{T} \rangle$; where $\overline{Q_{lh}}$ is the climatological latent heat flux, and \hat{T} is the perturbation of sea surface temperature from the mean and $\alpha = 0.067 K^{-1}$). The complete formulation of these separate equations that comprise G have been presented in the previous studies (Du et. al., 2008, Xie et. al., 2010). Q_{lh}^w is associated with changes in wind forcing and represents the effect of atmospheric forcing on SST via Q_{lh} , whereas Q_{lh}^o is sustained through changes in SST anomalies, and indicates the ocean damping effects via Q_{lh} . An extended area of positive Q_{lh}^w is seen in the eastern equatorial Indian Ocean and AS (Figure R2a), which reflects the decrease in wind speed associated with reduced latent heat cooling, as has been shown in the manuscript (Main Figure 1b). Furthermore, the evaporative damping in the equatorial eastern Indian Ocean shows a relatively weaker damping effect on SST anomalies relative to what is found for other regions in the Indian Ocean (Figure R2b), with this mainly due to weak surface winds. This is in fact the reason for why we chose to formally define and apply the shortwave-SST feedback (G pattern), namely in order to understand the initial surface warming response.

- *Du, Y. & Xie, S.-P. Role of atmospheric adjustments in the tropical Indian Ocean warming during the 20th century in climate models. Geophys. Res. Lett. 35, L08712 (2008).*

- Xie, S.-P. et al. Global Warming Pattern Formation: Sea Surface Temperature and Rainfall*. J. Clim. 23, 966–986 (2010).

Figure R2. Present-day decomposition of latent heat processes in the Indian Ocean. Ensemble mean of (a) the Q_{lh}^w atmospheric wind effect on the latent heat flux and (b) the Q_{lh}^o oceanic response of SST anomalies for the 50 ensemble members over the period 1980-2000. The equation used to calculate the atmospheric and oceanic effects is highlighted above in the response to the reviewer.

5. Lines 131-132: EIO has deep mean thermocline due to the warm pool there, which can be found in Fig 1e. Why do you say “a shallow mean thermocline” here?

Thank you for pointing this out. The text has now been revised in the new version of manuscript to say: “Projected easterly equatorial wind anomalies shoal the thermocline in the EIO and thereby generate anomalous upwelling of colder subsurface waters. This in turn contributes to offsetting some of the anthropogenic warming”

6. Lines 184-185: The effect of ITF in the southern IO has been revealed in Feng et al. (2011) and Dong and McPhaden (2016). Please cite the related papers.

Feng, M., C. Böning, A. Biastoch, E. Behrens, E. Weller, and Y. Masumoto, 2011: The reversal of the multidecadal trends of the equatorial Pacific easterly winds, and the Indonesian

Throughflow and Leeuwin Current transports. *Geophys. Res. Lett.*, 38, L11604, doi:10.1029/2011GL047291

Dong, L., and M. J. McPhaden, 2016: Interhemispheric SST gradient trends in the Indian Ocean prior to and during the recent global warming hiatus. *Journal of Climate*, 29(24), 9077-9095, <https://doi.org/10.1175/JCLI-D-16-0130.1>

Thank you for the suggestion. Both the papers have now been cited in the revised version of manuscript.

7. Line 196: To support your statement, please specify the correlation coefficient between the Qsw and SST over the SEIO and its statistical significance.

Thank you for the suggestion. The correlation coefficient between Qsw and SST over the SEIO and its statistical significance have now been included in the revised version.

8. Line 209: Figure 4 occurred at the first time here as Fig. 4d, which is unusual. First occurred in the manuscript should be Fig. 4a. Please re-order the sub-figures of Figure 4.

Thank you for pointing this out. It has now been corrected in the revised version.

REVIEWER COMMENTS

Reviewer #1 (Remarks to the Author):

Manuscript Title: Future Indian Ocean warming patterns

The author made a lot of effort to revise the manuscript. After reviewing this paper again, I find a few points that the author should address.

Major comments:

1.The future TIO warming pattern is not compared with the past 150 years. Based on the previous study, both of them seem much different from each other. The past warming pattern was regarded as the joint effect of the monsoon system and global climate coherence. This point is not investigated and discussed in the present version.

2.The author believes that the shortwave feedback is the key factor leading to less EIO warming. But as mentioned in Lines 135-139 of the revised manuscript, there are other processes involved. The author may use a similar Figure as Fig.2 and 3 to illustrate those processes.

3.This research concludes that G is responsible for a less-warm EIO, which further causes the equatorial easterly anomalies. Meanwhile, the easterly anomalies can be seen in the uniform 4K warming AMIP simulations, suggesting that the SST gradient is unnecessary for forming easterly anomalies along the IO equator. How to interpret this?

4.For the model bias, previous modeling study show TIO has a large bias in the mean status. I saw some remarkable difference between the CESM modeled and observed historical IO warming patterns in Fig R5: e.g., the cold anomalies along the Sumatra coast is missing in the observations. Considering it is a key region discussed in this paper, more analyses may be needed to explain this discrepancy. The author may also need to show the present-day G pattern in OBS aligned with Fig1(b).

5.Line 225: why does the low-level southern hemisphere cyclonic circulation cause the reduction in the low cloud cover? According to Fig3(e), there is a low-pressure system presented.

Minor comments:

1.Line 129 "Supplementary Fig 3": is it the right Figure?

2.Fig.2 shows AS on the left, while Fig. 3 puts AS on the right. Suggest re-order the panels to maintain consistency.

Reviewer #2 (Remarks to the Author):

I appreciate the authors' efforts in addressing my previous comments, and I find most of the responses satisfactory. The manuscript also has improved. However, I still have one major concern about the cause for weakening of the Indian Ocean Walker circulation.

By comparing the AMIP experiments from CMIP6 models, the authors conclude that both uniform warming and SST warming pattern may cause easterly wind anomalies over the equatorial Indian Ocean, with the former playing a more important role. However, it seems that the Indian Ocean wind anomalies in Figure 5a are much weaker than 5b, and the the associated wind anomalies in Figure 5a are not statistically significant. Hence, it seems that the weakened Indian Ocean Walker circulation is indeed caused by remote forcing such as El Niño warming. This may also affect the conclusion that the negative cloud-SST feedback causes the eastern Indian Ocean cooling.

Reviewer #3 (Remarks to the Author):

I am satisfied with the revisions and don't have more comments. Thanks for your effort in emphasizing my comments.

Reply to two reviewers' comments

We are grateful to the reviewers for revisiting our manuscript and for having again provided constructive comments and suggestions. In this document, we submit a point-by-point reply to each of the questions and requests for clarification that were raised. To facilitate reading of this document, reviewer comments are shown in black *italic* and our replies are marked in blue.

Reviewer #1 (Remarks to the Author):

Manuscript Title: Future Indian Ocean warming patterns

The author made a lot of effort to revise the manuscript. After reviewing this paper again, I find a few points that the author should address.

We sincerely value the reviewer's insightful comments and thorough evaluation of the revised version of the manuscript. In the revised version of the manuscript, we have addressed all of the reviewer's concerns, and our responses are provided below.

Major comments:

1. The future TIO warming pattern is not compared with the past 150 years. Based on the previous study, both of them seem much different from each other. The past warming pattern was regarded as the joint effect of the monsoon system and global climate coherence. This point is not investigated and discussed in the present version.

Previous studies have shown that the changes in monsoon circulation influence the historical SST warming patterns over the Indian Ocean (Swapna *et al.*, 2014). In Fig. R1 of this document (see below), we have compared the pattern of future projected Indian Ocean warming with the pattern of observed historical changes. Our analysis using the observational dataset showed a large warming maximum over the western equatorial Indian Ocean and the Arabian Sea. Additionally, another local maximum can be identified to the south of 20°S in the Indian Ocean. In contrast, the future projected warming pattern, with enhanced warming over the Arabian Sea

and south-eastern Indian Ocean (already discussed in the manuscript), is distinct from the historical warming pattern. As pointed out by the reviewer, this difference arises because the past warming pattern was shaped mainly by the joint effect natural climate variability and anthropogenic forcings (Swapna *et al.*, 2014, Wang *et al.*, 2017). This point has been addressed in the revised version of the manuscript (line 109-115).

- Swapna, P., Krishnan, R. & Wallace, J. M. Indian Ocean and monsoon coupled interactions in a warming environment. *Clim. Dyn.* **42**, 2439–2454 (2014).
- Wang, P. X. *et al.* The global monsoon across time scales: Mechanisms and outstanding issues. *Earth-Sci. Rev.* **174**, 84–121 (2017).

Figure R1. Comparison of historical and future Indian Ocean warming patterns. (a) Observed historical SST changes based on the HadISST dataset. The changes are calculated by taking the difference between the two periods (1880-1920 minus 1980-2020). **(b)** Ensemble mean changes in SST for 50 members of CESM2-LE. The mean state changes are calculated by taking the difference between the future (2060-2100) and the historical (1980-2020) periods. Note that different color scales are used between the plots to highlight the spatial patterns.

2. The author believes that the shortwave feedback is the key factor leading to less EIO warming. But as mentioned in Lines 135-139 of the revised manuscript, there are other processes involved. The author may use a similar Figure as Fig.2 and 3 to illustrate those processes. We have now added a figure in the Supplementary Materials (Fig. R2 below) to elucidate the roles of other processes in the reduced EIO warming in the future. Our analysis indicates that advection, especially zonal advection, among these processes plays a major role in reducing anomalous warming over the EIO (Fig. R2a). The projected easterly equatorial wind anomalies are associated with a shoaling of the thermocline in the EIO and an anomalous upwelling of colder subsurface waters there. This set of processes serves to mute or diminish the amplitude of anthropogenic warming there, as was described in the main body of the manuscript. Additionally, changes in long wave fluxes associated with decreases in mid- and high-level clouds lead to reduced EIO warming in the future (Fig. R2b).

Figure R2 Contribution of different factors to the reduced degree of warming over the Eastern Indian Ocean. Time series of each term of the vertically-integrated (0-65m) ocean heat budget equation over (a) the EIO (85°E:105°E, 10°S:5°N) from 1850-2100 for 50 ensemble members of the CESM2-LE. Inset plot (i) in (a) shows the time series of each component of the surface air-sea flux (positive values represent anomalous heat input into the ocean) components including shortwave radiation (Q_{sw}), longwave radiation (Q_{lw}), sensible

heat flux (Q_{sh}), latent heat flux (Q_{lh}), and net surface air-sea flux (Q_{net}) and **(ii)** represents the time series of vertically integrated individual advection terms such as zonal (X_{adv}), meridional (Y_{adv}) and vertical (Z_{adv}) advection. **(b)** Time series of vertically-integrated cloud cover over the EIO from 1850-2100 for 50 ensemble members. Inset plot **(iii)** in **(b)** shows the time series of the longwave cloud forcing (LWCF), shortwave cloud forcing (SWCF), and net (LWCF+SWCF) cloud forcing, and **(iv)** represents the changes in cloud liquid amount, cloud ice amount and cloud fraction over the EIO from the surface to 300hPa. Note that the IO mean is removed from each area-averaged time series over the EIO. All the time series are relative to an 1850-2014 baseline, and finally, 15-year running means are calculated. Thick (thin) lines represent the ensemble mean (individual ensemble members) in all the plots.

3. This research concludes that G is responsible for a less-warm EIO, which further causes the equatorial easterly anomalies. Meanwhile, the easterly anomalies can be seen in the uniform 4K warming AMIP simulations, suggesting that the SST gradient is unnecessary for forming easterly anomalies along the IO equator. How to interpret this?

This is indeed an interesting point. To clarify this ambiguity, we have compared the surface easterly wind stress anomalies over the equatorial central Indian Ocean between the uniform 4-K warming AMIP simulations and the coupled CESM2-LE simulations. As shown in Fig. R3, the magnitude of easterly wind anomalies are smaller in the uniform warming simulations than in the CESM2-LE. This implies that changes in zonal SST gradient also play a role in forming easterly anomalies anomaly the IO equator.

Figure R3. Response of the surface zonal wind over the equatorial central Indian Ocean to global warming. The bar plots show the surface zonal wind stress anomaly over the equatorial central Indian Ocean per unit global warming for the uniform 4-K warming AMIP experiment and the CESM2 LE. For the uniform AMIP simulations, the multi-model mean change is computed using 7 models (detailed in the Methods section of the manuscript). In the case of the CESM2 LE, the changes are calculated by taking difference between the future (2060-2100) and historical (1980-2020) periods, respectively, for 50 ensemble members and then the ensemble mean is computed.

4. For the model bias, previous modeling study show TIO has a large bias in the mean status. I saw some remarkable difference between the CESM modeled and observed historical IO warming patterns in Fig R5: e.g., the cold anomalies along the Sumatra coast is missing in the observations. Considering it is a key region discussed in this paper, more analyses may be needed to explain this discrepancy. The author may also need to show the present-day G pattern in OBS aligned with Fig1(b).

It seems that the use of different color scales may have caused some confusion. Figure R4 again shows the historical IO warming patterns in both observations and model simulations obtained from an EOF analysis over 1950-2010 for the subdomain defined by 20°N:20°S, 40°E:120°E. As shown in a previous study (Chu et al., 2014), the first two leading EOF modes are

characterized, respectively, by basin-wide warming in the Indian Ocean (first mode, Fig. R4 a, b) and a zonal dipole mode (second mode, Fig. R4 c, d) in both observations and the CESM2 LE.

In the revised version of the manuscript, we also include the present-day (historical) G pattern identified using the observations (green contour lines in Main Figure 1b, Figure R5). As in the CESM2 LE simulations, negative G values are found over the eastern equatorial Indian Ocean, indicating strong damping of SST anomalies.

- *Chu, J.-E. et al. Future change of the Indian Ocean basin-wide and dipole modes in the CMIP5. Clim. Dyn. 43, 535–551 (2014).*

Figure R4. Historical warming pattern in the Indian Ocean. The upper panel shows the Empirical orthogonal function (EOF-1) of the annual mean SST of (a) HadISST and (b) the ensemble mean of 50 members of CESM2-LE over the Indian Ocean (20°N:20°S, 40°E:120°E) over the period 1950 to 2010. (c and d) are the same as (a and b) but the EOF-2. A linear trend is removed from the dataset, and anomalies are calculated relative to the 1960-1990 baseline period.

Figure R5. G from observations for the historical period. The plot shows the G pattern during the period 1980-2000 (similar to the CESM2-LE simulation in the manuscript). G is multiplied by the factor of 10^3 and the anomalies are relative to the 1870-2014 baseline period. HadISST is used for the SST dataset and the NOAA- 20th Century Reanalysis V3 for the downward solar radiative flux.

5.Line 225: why does the low-level southern hemisphere cyclonic circulation cause the reduction in the low cloud cover? According to Fig3(e), there is a low-pressure system presented.

We now provide the explanation of this in the revised manuscript (line 239-241) “*In addition, the reduction in low-level clouds is likely to be accompanied by the vertical motion of an anomalous cyclonic circulation (Fig. 1a), which is associated with an increase in mid-level clouds (Fig. 3b) and the weakening of downward motion over the SEIO*”

Minor comments:

1.Line 129 “Supplementary Fig 3”: is it the right Figure?

Corrected.

2.Fig.2 shows AS on the left, while Fig. 3 puts AS on the right. Suggest re-order the panels to maintain consistency.

Thank you for pointing out this. It is now corrected in the revised version of the manuscript.

Reviewer #2 (Remarks to the Author):

I appreciate the authors' efforts in addressing my previous comments, and I find most of the responses satisfactory. The manuscript also has improved. However, I still have one major concern about the cause for weakening of the Indian Ocean Walker circulation. By comparing the AMIP experiments from CMIP6 models, the authors conclude that both uniform warming and SST warming pattern may cause easterly wind anomalies over the equatorial Indian Ocean, with the former playing a more important role. However, it seems that the Indian Ocean wind anomalies in Figure 5a are much weaker than 5b, and the associated wind anomalies in Figure 5a are not statistically significant. Hence, it seems that the weakened Indian Ocean Walker circulation is indeed caused by remote forcing such as El Niño warming. This may also affect the conclusion that the negative cloud-SST feedback causes the eastern Indian Ocean cooling.

Thank you for re-reviewing our manuscript and providing constructive comments and suggestions for improving the manuscript. We really appreciate that the point raised here shows the need for us to provide further clarification. Evidence from previous studies has demonstrated that El Niño-related atmospheric circulation perturbations induce SST perturbations over the Indian Ocean by modulating the surface air-sea fluxes and the depth of the oceanic mixed layer (*Villwock and Latif 1994, Venzke et al., 2000, Lau et al., 2003*), and that the El Niño signal in the Pacific also sustains a weakening of the ascending motion of the atmosphere over the eastern Indian Ocean through the Walker Circulation. These perturbations can in turn lead to a reorganization of surface winds and SST anomalies over the equatorial Indian Ocean (*Klein et al., 1998, Wang 2019*). We can then use these relationships as a first order analogue to understand the potential teleconnections from an El Niño-like warming pattern to the Indian Ocean region. Moreover, the local forcing in the Indian Ocean, such as a pIOD-like warming pattern, also results in the weakening of ascending motion over the eastern

Indian Ocean (*Cai et al., 2014*) and implies the slowdown of the Indian Ocean Walker Circulation. This indicates that the remote forcing, along with local forcing in the Indian Ocean, is likely to affect the Indian Ocean Walker circulation in the future climate, as pointed out by the reviewer.

Our results using the uniform warming and pattern warming AMIP experiments represent a weakening of the Walker circulation, with the magnitude of anomalous descending motion over the eastern Indian Ocean is greater with uniform warming than it is with pattern warming. Nevertheless, both sets of experiments project a strengthening of anomalous easterlies over the equatorial central Indian Ocean, with the magnitude greater with pattern warming than with uniform warming (Main Figure 5). This apparent discrepancy appears to indicate that other factors also control the magnitude of the anomalous descending motion over the eastern Indian Ocean or the weakening of the Indian Ocean Walker Circulation under future climate change. Despite these uncertainties, previous studies have shown that the Indian Ocean Walker Circulation can be weakened without the changes in the SST spatial pattern (*Ma et al., 2012, Sharma et al., 2022*). More specifically, they have demonstrated that although the projected weakening of the Indian Ocean Walker circulation is partly due to the pIOD-like warming pattern or the El Niño-like warming pattern, the local component in the Indian Ocean, such as the local meridional circulation, which is independent of the SST spatial pattern, also plays a crucial role by causing a strong anomalous descending motion over the Indian Ocean. This provides strong support for our assertion that remote forcing might not be the leading cause of the projected weakening of the Indian Ocean Walker Circulation.

- *Villwock, A., and M. Latif, 1994: Indian Ocean response to ENSO. Proc. Int. Conf. on Monsoon Variability and Prediction, Vol. II, Geneva, Switzerland, World Meteor. Org., 530–537.*
- *Klein, S. A., B. J. Soden, and N. C. Lau, 1999: Remote sea surface temperature variations during ENSO: Evidence for a tropical atmospheric bridge. J. Climate, 12, 917–932.*

- Venzke, S., M. Latif, and A. Villwock, 2000: *The coupled GCM ECHO-2. Part II: Indian Ocean response to ENSO*. *J. Climate*, **13**, 1371–1383.
- Lau, N.-C., and M. J. Nath, 2003: *Atmosphere-ocean variations in the Indo-Pacific sector during ENSO episodes*. *J. Climate*, **16**, 3–20.
- Wang, C *Three-ocean interaction and climate variability: a review and perspective*. *Clim Dyn* 53, 5119-5136. <https://doi.org/10.1007/s00382-019-04930-x>
- Cai, W. et al. *Projected response of the Indian Ocean dipole to greenhouse warming*. *Nat. Geosci.* **6**, 999–1007 (2013).
- Ma, J., Xie, S.-P., & Kosaka, Y. (2012). *Mechanisms for Tropical Tropospheric Circulation Change in Response to Global Warming*. *Journal of Climate*, 25(8), 2979–2994. <http://www.jstor.org/stable/26191363>
- Sharma, S., Ha, KJ., Cai, W. et al. *Local meridional circulation changes contribute to a projected slowdown of the Indian Ocean Walker circulation*. *npj Clim Atmos Sci* **5**, 15 (2022). <https://doi.org/10.1038/s41612-022-00242-w>

Reviewer #3 (Remarks to the Author):

I am satisfied with the revisions and don't have more comments. Thanks for your effort in emphasizing my comments.

Thank you for reviewing our manuscript and spending considerable time providing detailed comments which have helped us to further clarify and improve the manuscript.

REVIEWERS' COMMENTS

Reviewer #1 (Remarks to the Author):

This revised manuscript is much improved, and the authors have thoughtfully responded to my previous comments.

I think the paper is approaching a form acceptable for publication.

Reviewer #2 (Remarks to the Author):

I am glad that my concern has been addressed satisfactorily, and I recommend acceptance of the manuscript for publication.